# Blending MPC & Value Function Approximation for Efficient Reinforcement Learning

**Mohak Bhardwaj**[1]    **Sanjiban Choudhury**[2]    **Byron Boots**[1]

[1] University of Washington    [2] Aurora Innovation Inc.

## Abstract

Model-Predictive Control (MPC) is a powerful tool for controlling complex, real-world systems that uses a model to make predictions about future behavior. For each state encountered, MPC solves an online optimization problem to choose a control action that will minimize future cost. This is a surprisingly effective strategy, but real-time performance requirements warrant the use of simple models. If the model is not sufficiently accurate, then the resulting controller can be biased, limiting performance. We present a framework for improving on MPC with model-free reinforcement learning (RL). The key insight is to view MPC as constructing a series of local Q-function approximations. We show that by using a parameter $\lambda$, similar to the trace decay parameter in TD($\lambda$), we can systematically trade-off learned value estimates against the local Q-function approximations. We present a theoretical analysis that shows how error from inaccurate models in MPC and value function estimation in RL can be balanced. We further propose an algorithm that changes $\lambda$ over time to reduce the dependence on MPC as our estimates of the value function improve, and test the efficacy our approach on challenging high-dimensional manipulation tasks with biased models in simulation. We demonstrate that our approach can obtain performance comparable with MPC with access to true dynamics even under severe model bias and is more sample efficient as compared to model-free RL.

## 1 Introduction

Model-free Reinforcement Learning (RL) is increasingly used in challenging sequential decision-making problems including high-dimensional robotics control tasks (Haarnoja et al., 2018; Schulman et al., 2017) as well as video and board games (Silver et al., 2016; 2017). While these approaches are extremely general, and can theoretically solve complex problems with little prior knowledge, they also typically require a large quantity of training data to succeed. In robotics and engineering domains, data may be collected from real-world interaction, a process that can be dangerous, time consuming, and expensive.

Model-Predictive Control (MPC) offers a simpler, more practical alternative. While RL typically uses data to learn a global model offline, which is then deployed at test time, MPC solves for a policy *online* by optimizing an approximate model for a finite horizon at a given state. This policy is then executed for a single timestep and the process repeats. MPC is one of the most popular approaches for control of complex, safety-critical systems such as autonomous helicopters (Abbeel et al., 2010), aggressive off-road vehicles (Williams et al., 2016) and humanoid robots (Erez et al., 2013), owing to its ability to use approximate models to optimize complex cost functions with nonlinear constraints (Mayne et al., 2000; 2011).

However, approximations in the model used by MPC can significantly limit performance. Specifically, *model bias* may result in *persistent errors* that eventually compound and become catastrophic. For example, in non-prehensile manipulation, practitioners often use a simple quasi-static model that assumes an object does not roll or slide away when pushed. For more dynamic objects, this can lead to aggressive pushing policies that perpetually over-correct, eventually driving the object off the surface.

Recently, there have been several attempts to combine MPC with model free RL, showing that the combination can improve over the individual approaches alone. Many of these approaches involve using RL to learn a terminal cost function, thereby increasing the effective horizon of MPC (Zhong et al., 2013; Lowrey et al., 2018; Bhardwaj et al., 2020). However, the learned value function is only applied at the end of the MPC horizon. Model errors would still persist in horizon, leading to sub-optimal policies. Similar approaches have also been applied to great effect in discrete games with known models (Silver et al., 2016; 2017; Anthony et al., 2017), where value functions and policies learned via model-free RL are used to

guide Monte-Carlo Tree Search. In this paper, we focus on a somewhat broader question: can machine learning be used to both increase the effective horizon of MPC, while also correcting for model bias?

One straightforward approach is to try to learn (or correct) the MPC model from real data encountered during execution; however there are some practical barriers to this strategy. Hand-constructed models are often crude-approximations of reality and lack the expressivity to represent encountered dynamics. Moreover, increasing the complexity of such models leads to computationally expensive updates that can harm MPC's online performance. Model-based RL approaches such as Chua et al. (2018); Nagabandi et al. (2018); Shyam et al. (2019) aim to learn general neural network models directly from data. However, learning globally consistent models is an exceptionally hard task due to issues such as covariate shift (Ross & Bagnell, 2012).

We propose a framework, MPQ($\lambda$), for weaving together MPC with learned value estimates to trade-off errors in the MPC model and approximation error in a learned value function. Our key insight is to view MPC as tracing out a series of local Q-function approximations. We can then blend each of these Q-functions with value estimates from reinforcement learning. We show that by using a blending parameter $\lambda$, similar to the trace decay parameter in TD($\lambda$), we can systematically trade-off errors between these two sources. Moreover, by smoothly decaying $\lambda$ over learning episodes we can achieve the best of both worlds - a policy can depend on a prior model before its has encountered any data and then gradually become more reliant on learned value estimates as it gains experience.

To summarize, our key contributions are:
1. A framework that unifies MPC and Model-free RL through value function approximation.
2. Theoretical analysis of finite horizon planning with approximate models and value functions.
3. Empirical evaluation on challenging manipulation problems with varying degrees of model-bias.

## 2 PRELIMINARIES

### 2.1 REINFORCEMENT LEARNING

We consider an agent acting in an infinite-horizon discounted Markov Decision Process (MDP). An MDP is defined by a tuple $\mathcal{M} = (\mathcal{S}, \mathcal{A}, c, P, \gamma, \mu)$ where $\mathcal{S}$ is the state space, $\mathcal{A}$ is the action space, $c(s,a)$ is the per-step cost function, $s_{t+1} \sim P(\cdot | s_t, a_t)$ is the stochastic transition dynamics and $\gamma$ is the discount factor and $\mu(s_0)$ is a distribution over initial states. A closed-loop policy $\pi(\cdot | s)$ outputs a distribution over actions given a state. Let $\mu_{\mathcal{M}}^{\pi}$ be the distribution over state-action trajectories obtained by running policy $\pi$ on $\mathcal{M}$. The value function for a given policy $\pi$, is defined as $V_{\mathcal{M}}^{\pi}(s) = \mathbb{E}_{\mu_{\mathcal{M}}^{\pi}} \left[ \sum_{t=0}^{\infty} \gamma^t c(s_t, a_t) | s_0 = s \right]$ and the action-value function as $Q_{\mathcal{M}}^{\pi}(s,a) = \mathbb{E}_{\mu_{\mathcal{M}}^{\pi}} \left[ \sum_{t=0}^{\infty} \gamma^t c(s_t, a_t) | s_0 = s, a_0 = a \right]$. The objective is to find an optimal policy $\pi^* = \underset{\pi}{\mathrm{argmin}} \, \mathbb{E}_{s_0 \sim \mu} [V_{\mathcal{M}}^{\pi}(s_0)]$. We can also define the (dis)-advantage function $A_{\mathcal{M}}^{\pi}(s,a) = Q_{\mathcal{M}}^{\pi}(s,a) - V^{\pi}(s)$, which measures how good an action is compared to the action taken by the policy in expectation. It can be equivalently expressed in terms of the Bellman error as $A_{\mathcal{M}}^{\pi}(s,a) = c(s,a) + \gamma \mathbb{E}_{s' \sim P, a' \sim \pi}[Q_{\mathcal{M}}^{\pi}(s', a')] - \mathbb{E}_{a \sim \pi}[Q_{\mathcal{M}}^{\pi}(s,a)]$.

### 2.2 MODEL-PREDICTIVE CONTROL

MPC is a widely used technique for synthesizing closed-loop policies for MDPs. Instead of trying to solve for a single, globally optimal policy, MPC follows a more pragmatic approach of optimizing simple, local policies *online*. At every timestep on the system, MPC uses an approximate model of the environment to search for a parameterized policy that minimizes cost over a finite horizon. An action is sampled from the policy and executed on the system. The process is then repeated from the next state, often by warm-starting the optimization from the previous solution.

We formalize this process as solving a simpler *surrogate* MDP $\hat{\mathcal{M}} = (\mathcal{S}, \mathcal{A}, \hat{c}, \hat{P}, \gamma, \hat{\mu}, H)$ online, which differs from $\mathcal{M}$ by using an approximate cost function $\hat{c}$, transition dynamics $\hat{P}$ and limiting horizon to $H$. Since it plans to a finite horizon, it's also common to use a terminal state-action value function $\hat{Q}$ that estimates the cost-to-go. The start state distribution $\hat{\mu}$ is a dirac-delta function centered on the current state $s_0 = s_t$. MPC can be viewed as iteratively constructing an estimate of the Q-function of the original MDP $\mathcal{M}$, given policy $\pi_\phi$ at state $s$:

$$Q_H^\phi(s,a) = \mathbb{E}_{\mu_{\mathcal{M}}^{\pi_\phi}} \left[ \sum_{i=0}^{H-1} \gamma^i \hat{c}(s_i, a_i) + \gamma^H \hat{Q}(s_H, a_H) | s_0 = s, a_0 = a \right] \quad (1)$$

MPC then iteratively optimizes this estimate (at current system state $s_t$) to update the policy parameters

$$\phi_t^* = \operatorname*{argmin}_\phi Q_H^\phi(s_t, \pi_\phi(s_t)) \tag{2}$$

Alternatively, we can also view the above procedure from the perspective of disadvantage minimization. Let us define an estimator for the 1-step disadvantage with respect to the potential function $\hat{Q}$ as $A(s_i, a_i) = c(s_i, a_i) + \gamma \hat{Q}(s_{i+1}, a_{i+1}) - \hat{Q}(s_i, a_i)$. We can then equivalently write the above optimization as minimizing the discounted sum of disadvantages over time via the telescoping sum trick

$$\operatorname*{argmin}_{\pi \in \Pi} \mathbb{E}_{\mu_\mathcal{M}^{\pi_\phi}} \left[ \hat{Q}(s_0, a_0) + \sum_{i=0}^{H-1} \gamma^i A(s_i, a_i) \,|\, s_0 = s_t \right] \tag{3}$$

Although the above formulation queries the $\hat{Q}$ at every timestep, it is still exactly equivalent to the original problem and hence, does not mitigate the effects of model-bias. In the next section, we build a concrete method to address this issue by formulating a novel way to blend Q-estimates from MPC and a learned value function that can balance their respective errors.

## 3 MITIGATING BIAS IN MPC VIA REINFORCEMENT LEARNING

In this section, we develop our approach to systematically deal with model bias in MPC by blending-in learned value estimates. First, we take a closer look at the different sources of error in the estimate in (1) and then propose an easy-to-implement, yet effective strategy for trading them off.

### 3.1 SOURCES OF ERROR IN MPC

The performance of MPC algorithms critically depends on the quality of the Q-function estimator $Q_H^\phi(s, a)$ in (1). There are three major sources of approximation error. First, model bias can cause compounding errors in predicted state trajectories, which biases the estimates of the costs of different action sequences. The effect of model error becomes more severe as $H \to \infty$. Second, the error in the terminal value function gets propagated back to the estimate of the Q-function at the start state. With discounting, the effect of error due to inaccurate terminal value function diminishes as $H$ increases. Third, using a small $H$ with an inaccurate terminal value function can make the MPC algorithm greedy and myopic to rewards further out in the future.

We can formally bound the performance of the policy with approximate models and approximate learned value functions. In Theorem 3.1, we show the loss in performance of the resulting policy as a function of the model error, value function error and the planning horizon.

**Theorem 3.1** (Proof Appendix A.1.2)**.** *Let MDP $\hat{\mathcal{M}}$ be an $\alpha$-approximation of $\mathcal{M}$ such that $\forall (s, a)$, we have $\left\| \hat{P}(s'|s, a) - P(s'|s, a) \right\|_1 \leq \alpha$ and $|\hat{c}(s, a) - c(s, a)| \leq \alpha$. Let the learned value function $\hat{Q}(s, a)$ be an $\epsilon$-approximation of the true value function $\left\| \hat{Q}(s, a) - Q_\mathcal{M}^{\pi^*}(s, a) \right\|_\infty \leq \epsilon$. The performance of the MPC policy is bounded w.r.t the optimal policy as $\left\| V_\mathcal{M}^{\pi^*}(s) - V_{\hat{\mathcal{M}}}^{\hat{\pi}}(s) \right\|_\infty$*

$$\leq 2 \left( \frac{\gamma(1 - \gamma^{H-1})}{(1 - \gamma^H)(1 - \gamma)} \alpha H \left( \frac{c_{\max} - c_{\min}}{2} \right) + \frac{\gamma^H \alpha H}{1 - \gamma^H} \left( \frac{V_{\max} - V_{\min}}{2} \right) + \frac{\alpha}{1 - \gamma} + \frac{\gamma^H \epsilon}{1 - \gamma^H} \right) \tag{4}$$

This theorem generalizes over various established results. Setting $H = 1, \epsilon = 0$ gives us the 1-step simulation lemma in Kearns & Singh (2002) (Appendix A.1.1). Setting $\alpha = 0$, i.e. true model, recovers the cost-shaping result in Sun et al. (2018). Further inspecting terms in (4), we see that the model error *increases* with horizon $H$ (the first two terms) while the learned value error *decreases* with $H$ which matches our intuitions.

In practice, the errors in model and value function are usually unknown and hard to estimate making it impossible to set the MPC horizon to the optimal value. Instead, we next propose a strategy to blend the Q-estimates from MPC and the learned value function at every timestep along the horizon, instead of just the terminal step such that we can properly balance the different sources of error.

## 3.2 BLENDING MODEL PREDICTIVE CONTROL AND VALUE FUNCTIONS

A naive way to blend Q-estimates from MPC with Q-estimates from the value function would be to consider a convex combination of the two

$$(1-\lambda)\underbrace{\hat{Q}(s,a)}_{\text{model-free}}+\lambda\underbrace{Q_H^\phi(s,a)}_{\text{model-based}} \tag{5}$$

where $\lambda \in [0,1]$. Here, the value function is contributing to a residual that is added to the MPC output, an approach commonly used to combine model-based and model-free methods (Lee et al., 2020). However, this is solution is rather *ad hoc*. If we have at our disposal a value function, why invoke it at only at the first and last timestep? As the value function gets better, it should be useful to invoke it *at all timesteps*.

Instead, consider the following recursive formulation for the Q-estimate. Given $(s_i,a_i)$, the state-action encountered at horizon $i$, the blended estimate $Q^\lambda(s_i,a_i)$ is expressed as

$$\underbrace{Q^\lambda(s_i,a_i)}_{\text{current blended estimate}}=(1-\lambda)\underbrace{\hat{Q}(s_i,a_i)}_{\text{model-free}}+\lambda(\underbrace{\hat{c}(s_i,a_i)}_{\text{model-based}}+\gamma\underbrace{Q^\lambda(s_{i+1},a_{i+1})}_{\text{future blended estimate}}) \tag{6}$$

where $\lambda \in [0,1]$. The recursion ends at $Q^\lambda(s_H,a_H)=\hat{Q}(s_H,a_H)$. In other words, the current blended estimate is a convex combination of the model-free value function and the one-step model-based return. The return in turn uses the future blended estimate. Note unlike (5), the model-free estimate is invoked at *every timestep*.

We can unroll (6) in time to show $Q_H^\lambda(s,a)$, the blended $H-$horizon estimate, is simply an exponentially weighted average of *all horizon* estimates

$$Q_H^\lambda(s,a)=(1-\lambda)\sum_{i=0}^{H-1}\lambda^i Q_i^\phi(s,a)+\lambda^H Q_H^\phi(s,a) \tag{7}$$

where $Q_k^\phi(s,a)=\mathbb{E}_{\mu_\mathcal{M}^{\pi_\phi}}\left[\sum_{i=0}^{k-1}\gamma^i\hat{c}(s_i,a_i)+\gamma^k\hat{Q}(s_k,a_k)\,|\,s_0=s,a_0=a\right]$ is a $k$-horizon estimate. When $\lambda=0$, the estimator reduces to the just using $\hat{Q}$ and when $\lambda=1$ we recover the original MPC estimate $Q_H$ in (1). For intermediate values of $\lambda$, we interpolate smoothly between the two by interpolating all $H$ estimates.

Implementing (7) naively would require running $H$ versions of MPC and then combining their outputs. This is far too expensive. However, we can switch to the disadvantage formulation by applying a similar telescoping trick

$$Q_H^\lambda(s,a)=\mathbb{E}_{\mu_\mathcal{M}^{\pi_\phi}}\left[\hat{Q}(s_0,a_0)+\sum_{i=0}^{H-1}(\gamma\lambda)^i A(s_i,a_i)\right] \tag{8}$$

This estimator has a similar form as the $TD(\lambda)$ estimator for the value function. However, while $TD(\lambda)$ uses the $\lambda$ parameter for bias-variance trade-off, our blended estimator aims trade-off bias in dynamics model with bias in learned value function.

Why use blending $\lambda$ when one can simply tune the horizon $H$? First, $H$ limits the *resolution* we can tune since it's an integer – as $H$ gets smaller the resolution becomes worse. Second, the blended estimator $Q_H^\lambda(s,a)$ uses far more samples. Say we have access to optimal horizon $H^*$. Even if both $Q_H^\lambda$ and $Q_{H^*}^\phi$ had the same bias, the latter uses a strict subset of samples as the former. Hence the variance of the blended estimator will be lower, with high probability.

## 4 THE MPQ($\lambda$) ALGORITHM

We develop a simple variant of Q-Learning, called Model-Predictive Q-Learning with $\lambda$ Weights (MPQ($\lambda$)) that learns a parameterized Q-function estimate $\hat{Q}_\theta$. Our algorithm, presented in Alg. 1, modifies Q-learning to use blended Q-estimates as described in the (8) for both action selection and generating value targets. The parameter $\lambda$ is used to trade-off the errors due to model-bias and learned Q-function, $\hat{Q}_\theta$. This can be viewed as an extension of the MPQ algorithm from Bhardwaj et al. (2020) to explicitly deal with model bias by incorporating the learned Q-function at all timesteps. Unlike MPQ, we do not explicitly consider the entropy-regularized formulation, although our framework can be modified to incorporate soft-Q targets.

---

**Algorithm 1:** MPQ($\lambda$)

---

**Input:** Initial Q-function weights $\theta$, Approximate dynamics $\hat{P}$ and cost function $\hat{c}$
**Parameter:** MPC horizon $H$, $\lambda$ schedule $[\lambda_1, \lambda_2, ...]$,
               discount factor $\gamma$, minibatch size $K$, num mini-batches $N$, update frequency $t_{update}$

1   $\mathcal{D} \leftarrow \emptyset$
2   **for** $t = 1...\infty$ **do**
      // Update $\lambda$
3     $\lambda = \lambda_t$
      // Blended MPC action selection
4     $\phi_t \leftarrow \underset{\phi}{\operatorname{argmin}} \mathbb{E}_{\mu_{\mathcal{M}}^{\pi_\phi}} \left[ \hat{Q}_\theta(s_0, a_0) + \sum_{i=0}^{H-1} (\gamma\lambda)^i A(s_i, a_i) \,|\, s_0 = s_t \right]$
5     $a_t \sim \pi_{\phi_t}$
6     Execute $a_t$ on the system and observe $(c_t, s_{t+1})$
7     $\mathcal{D} \leftarrow (s_t, a_t, c_t, s_{t+1})$
8     **if** $t \% t_{update} == 0$ **then**
9        Sample N minibatches $\left( \left\{ s_{k,n}, a_{k,n}, c_{k,n}, s'_{k,n} \right\}_{k=1}^K \right)_{n=1}^N$ from $\mathcal{D}$
         // Generate Blended MPC value targets
10       $\hat{y}_{k,n} = c_{k,n} + \gamma \min \mathbb{E}_{\mu_{\mathcal{M}}^{\pi_\phi}} \left[ \hat{Q}_\theta(s_0, a_0) + \sum_{i=0}^{H-1} (\gamma\lambda)^i A(s_i, a_i) \,|\, s_0 = s'_{k,n} \right]$
11       Update $\theta$ with SGD on loss $\mathcal{L} = \frac{1}{N} \frac{1}{K} \sum_{n=1}^N \sum_{k=1}^K \left( \hat{y}_{k,n} - \hat{Q}_\theta(s_{k,n}, a_{k,n}) \right)^2$

---

At every timestep $t$, MPQ($\lambda$) proceeds by using $H$-horizon MPC from the current state $s_t$ to optimize a policy $\pi_\phi$ with parameters $\phi$. We modify the MPC algorithm to optimize for the greedy policy with respect to the blended Q-estimator in (8), that is

$$\phi_t^* = \underset{\phi}{\operatorname{argmin}} \mathbb{E}_{\mu_{\mathcal{M}}^{\pi_\phi}} \left[ \hat{Q}_\theta(s_0, a_0) + \sum_{i=0}^{H-1} (\gamma\lambda)^i A(s_i, a_i) \,|\, s_0 = s_t \right] \tag{9}$$

An action sampled from the resulting policy is then executed on the system. A commonly used heuristic is to warm start the above optimization by shifting forward the solution from the previous timestep, which serves as a good initialization if the noise in the dynamics in small (Wagener et al., 2019). This can significantly cut computational cost by reducing the number of iterations required to optimize (9) at every timestep.

Periodically, the parameters $\theta$ are updated via stochastic gradient descent to minimize the following loss function with $N$ mini-batches of experience tuples of size $K$ sampled from the replay buffer

$$\mathcal{L}(\theta) = \frac{1}{N} \frac{1}{K} \sum_{n=1}^N \sum_{k=1}^K \left( \hat{y}_{k,n} - \hat{Q}_\theta(s_{k,n}, a_{k,n}) \right)^2 \tag{10}$$

The $H$-horizon MPC with blended Q-estimator is again invoked to calculate the targets

$$\hat{y}_j = c(s_j, a_j) + \gamma \min_\phi \mathbb{E}_{\mu_{\mathcal{M}}^{\pi_\phi}} \left[ \hat{Q}_\theta(s_0, a_0) + \sum_{i=0}^{H-1} (\gamma\lambda)^i A(s_i, a_i) \,|\, s_0 = s'_{k,n} \right] \tag{11}$$

Using MPC to reduce error in Q-targets has been previously explored in literature (Lowrey et al., 2018; Bhardwaj et al., 2020), where the model is either assumed to be perfect or model-error is not explicitly accounted for. MPC with the blended Q-estimator and an appropriate $\lambda$ allows us to generate more stable Q-targets than using $Q_\theta$ or model-based rollouts with a terminal Q-function alone. However, running H-horizon optimization for all samples in a mini-batch can be time-consuming, forcing the use of smaller batch sizes and sparse updates. In our experiments, we employ a practical modification where during

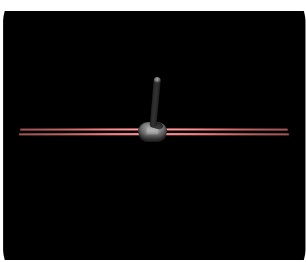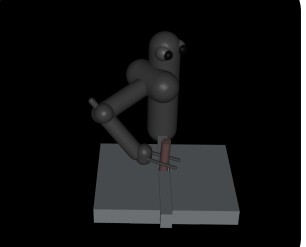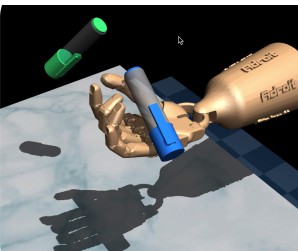

Figure 1: Tasks for evaluating MPQ($\lambda$). Left to right - cartpole, peg insertion with 7DOF arm, and in-hand manipulation to orient align pen(blue) with target(green) with 24DOF dexterous hand.

the action selection step, MPC is also queried for value targets which are then stored in the replay buffer, thus allowing us to use larger batch sizes and updates at every timestep.

Finally, we also allow $\lambda$ to vary over time. In practice, $\lambda$ is decayed as more data is collected on the system. Intuitively, in the early stages of learning, the bias in $\hat{Q}_\theta$ dominates and hence we want to rely more on the model. A larger value of $\lambda$ is appropriate as it up-weights longer horizon estimates in the blended-Q estimator. As $\hat{Q}_\theta$ estimates improve over time, a smaller $\lambda$ is favorable to reduce the reliance on the approximate model.

## 5 EXPERIMENTS

**Task Details:** We evaluate MPQ($\lambda$) on simulated robot control tasks, including a complex manipulation task with a 7DOF arm and in-hand manipulation with a 24DOF anthropomorphic hand (Rajeswaran* et al., 2018) as shown in Fig. 1. For each task, we provide the agent with a biased version of simulation that is used as the dynamics model for MPC. We use Model Predictive Path Integral Control (MPPI) (Williams et al., 2017), a state-of-the-art sampling-based algorithm as our MPC algorithm throughout.

1. CARTPOLESWINGUP: A classic control task where the agent slides a cart along a rail to swingup the pole attached via an unactuated hinge joint. Model bias is simulated by providing the agent incorrect masses of the cart and pole. The masses are set lower than the true values to make the problem harder for MPC as the algorithm will always input smaller controls than desired as also noted in Ramos et al. (2019). Initial position of cart and pole are randomized at every episode.

2. SAWYERPEGINSERTION: The agent controls a 7DOF Sawyer arm to insert a peg attached to the end-effector into a hole at different locations on a table in front of the robot. We test the effects of inaccurate perception by simulating a sensor at the target location that provides noisy position measurements at every timestep. MPC uses a deterministic model that does not take sensor noise into account as commonly done in controls. This biases the cost of simulated trajectories, causing MPC to fail to reach the target.

3. INHANDMANIPULATION: A challenging in-hand manipulation task with a 24DOF dexterous hand from Rajeswaran* et al. (2018). The agent must align the pen with target orientation within certain tolerance for succcess. The initial orientation of the pen is randomized at every episode. Here, we simulate bias by providing larger estimates of the mass and inertia of the pen as well as friction coefficients, which causes the MPC algorithm to optimize overly aggressive policies and drop the pen.

Please refer to the Appendix A.2 for more details of the tasks, success criterion and biased simulations.

**Baselines**: We compare MPQ($\lambda$) against both model-based and model-free baselines - MPPI with true dynamics and no value function, MPPI with biased dynamics and no value function and Proximal Policy Optimization (PPO) Schulman et al. (2017).

**Learning Details**: We represent the Q-function with a feed-forward neural network. We bias simulation parameters like mass or friction coefficients using the formula $m = (1+b)m_{true}$, where $b$ is a bias-factor. We also employ a practical modification to Alg. 1 in order to speed up training times as discussed in Section 4. Instead of maintaining a large replay-buffer and re-calculating targets for every experience tuple in a mini-batch, as done by approaches such as Bhardwaj et al. (2020); Lowrey et al. (2018), we simply query MPC for the value targets online and store them in a smaller buffer, which allows us to perform updates at every timestep. We use publicly the available implementation at https://bit.ly/38RcDrj for PPO. Refer to the Appendix A.2 for more details.

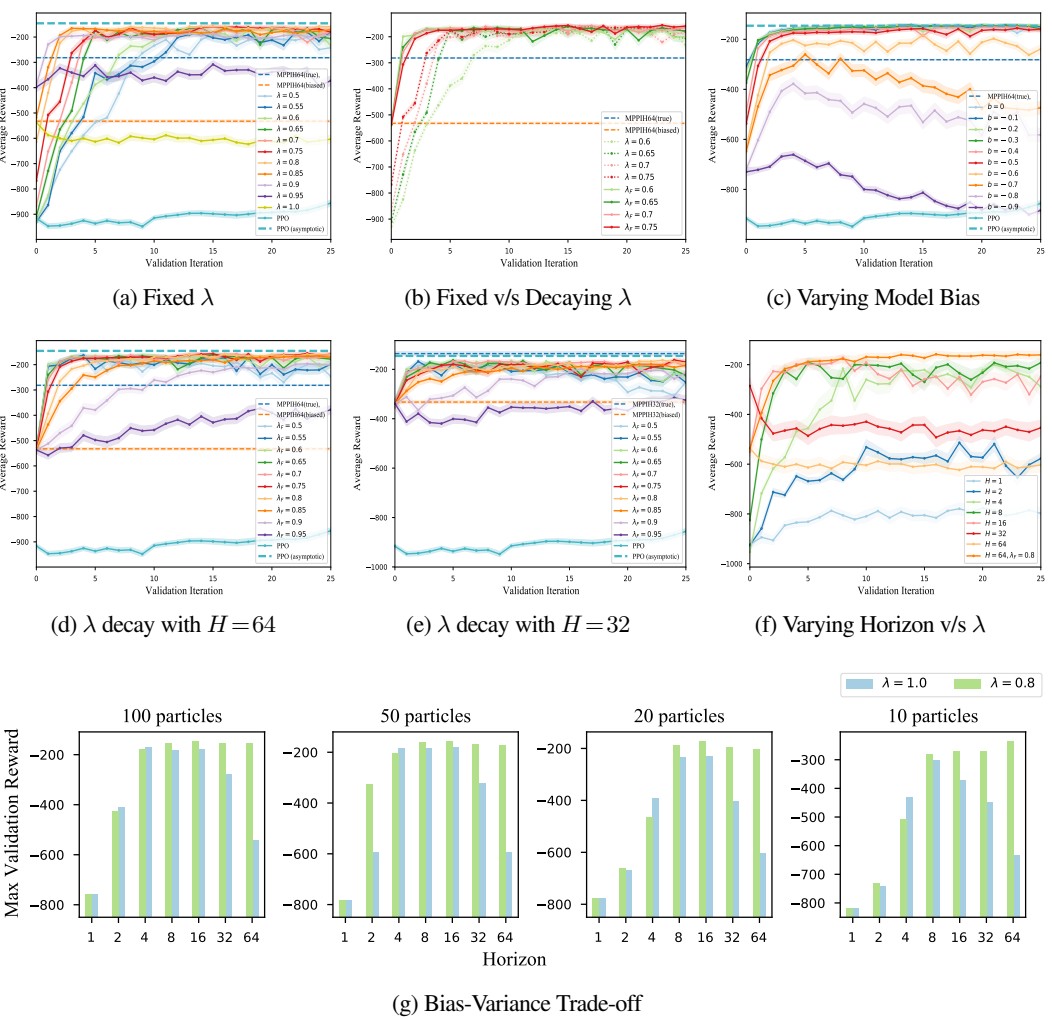

(a) Fixed $\lambda$     (b) Fixed v/s Decaying $\lambda$     (c) Varying Model Bias

(d) $\lambda$ decay with $H=64$     (e) $\lambda$ decay with $H=32$     (f) Varying Horizon v/s $\lambda$

(g) Bias-Variance Trade-off

Figure 2: CARTPOLESWINGUP experiments. Solid lines show average rewards over 30 validation episodes (fixed start states) with length of 100 steps and 3 runs with different seeds. The dashed lines are average reward of MPPI for the same validation episodes. Shaded region depicts the standard error of the mean that denotes the confidence on the average reward estimated from finite samples. Training is performed for 100k steps with validation after every 4k steps. When decaying $\lambda$ as per a schedule, it is fixed to the current value during validation. In (b),(d),(e), (f) $\lambda_F$ denotes the $\lambda$ value at the end of training. PPO asymptotic performance is reported as average reward of last 10 validation iterations. (g) shows the best validation reward at the end of training for different horizon values and MPPI trajectory samples (particles) using $\lambda=1.0$ and $\lambda=0.8$

## 5.1 ANALYSIS OF OVERALL PERFORMANCE

**O 1.** *MPQ($\lambda$) is able to overcome model-bias in MPC for a wide range of $\lambda$ values.*

Fig. 2(a) shows a comparison of MPQ($\lambda$) with MPPI using true and biased dynamics with $b=-0.5$ and $H=64$ for various settings of $\lambda$. There exists a wide range of $\lambda$ values for which MPQ($\lambda$) can efficiently trade-off model-bias with the bias in the learned Q-function and out-perform MPPI with biased dynamics. However, setting $\lambda$ to a high value of $1.0$ or $0.95$, which weighs longer horizons heavily leads to poor performance as compounding effects of model-bias are not compensated by $Q_\theta$. Performance also begins to drop as $\lambda$ decreases below $0.6$. MPQ($\lambda$) outperforms MPPI with access to the true dynamics and reaches close to asymptotic performance of PPO. This is not surprising as the learned Q-function adds global information to the optimization and $\lambda$ corrects for errors in optimizing longer horizons.

**O 2.** *Faster convergence can be achieved by decaying $\lambda$ over time.*

As more data is collected on the system, we expect the bias in $Q_\theta$ to decrease, whereas model bias remains constant. A larger value of $\lambda$ that favors longer horizons is better during initial steps of training as the effect of a randomly initialized $Q_\theta$ is diminished due to discounting and better exploration is achieved by forward lookahead. Conversely, as $Q_\theta$ gets more accurate, model-bias begins to hurt performance

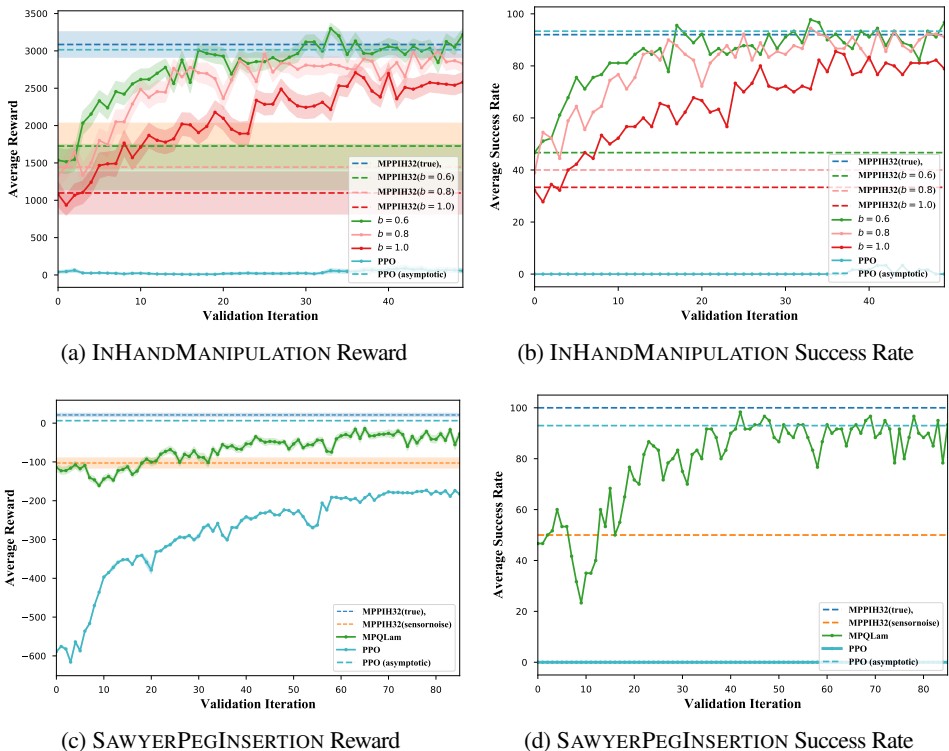

(a) INHANDMANIPULATION Reward

(b) INHANDMANIPULATION Success Rate

(c) SAWYERPEGINSERTION Reward

(d) SAWYERPEGINSERTION Success Rate

Figure 3: Robustness and sample efficiency of MPQ($\lambda$). (a),(b) Varying bias factor over mass, inertia and friction of pen (c),(d) Peg insertion with noisy perception. Total episode length is 75 steps for both. Same bias factor $b$ is used for all altered properties per task. Curves depict average reward over 30 validation episodes with multiple seeds and shaded areas are the standard error of the mean. Validation done after every 3k steps and $\lambda$ is decayed to 0.85 at end of 75k training steps in both. Asymptotic performance of PPO is average of last 10 validation iterations. Refer to Appendix A.2 for details on tasks and success metrics.

and a smaller $\lambda$ is favorable. We test this by decaying $\lambda$ in $[1.0, \lambda_F]$ using a fixed schedule and observe that indeed faster convergence is obtained by reducing the dependence on the model over training steps as shown in 2(b). Figures 2(d) and 2(e) present ablations that show that MPQ($\lambda$) is robust to a wide range of decay rates with $H = 64$ and 32 respectively. When provided with true dynamics, MPPI with $H = 32$ performs better than $H = 64$ due to optimization issues with long horizons. MPQ($\lambda$) reaches performance comparable with MPPI $H = 32$ and asymptotic performance of PPO in both cases showing robustness to horizon values which is important since in practice we wish to set the horizon as large as our computation budget permits. However, decaying $\lambda$ too fast or too slow can have adverse effects on performance. An interesting question for future work is whether $\lambda$ can be adapted in a state-dependent manner. Refer to Appendix A.2 for details on the decay schedule.

**O 3.** MPQ($\lambda$) *is much more sample efficient compared to model-free RL on high-dimensional continuous control tasks, while maintaining asymptotic performance.*

Figures 2 and 3 show a comparison of MPQ($\lambda$) with the model-free PPO baseline. In all cases, we observe that MPQ($\lambda$), through its use of approximate models, learned value functions, and a dynamically-varying $\lambda$ parameter to trade-off different sources of error, rapidly improves its performance and achieves average reward and success rate comparable to MPPI with access to ground truth dynamics and model-free RL in the limit. In INHANDMANIPULATION, PPO performance does not improve at all over 150k training steps. In SAWYERPEGINSERTION, the small magnitude of reward difference between MPPI with true and biased models is due to the fact that despite model bias, MPC is able to get the peg close to the table, but sensor noise inhibits precise control to consistently insert it in the hole. Here, the value function learned by MPQ($\lambda$) can adapt to sensor noise and allow for fine-grained control near the table.

**O 4.** MPQ($\lambda$) *is robust to large degree of model misspecification.*

Fig. 2(c) shows the effects of different values of the bias factor $b$ used to vary the mass of the cart and pole for MPQ($\lambda$) with a fixed $\lambda$ decay rate of $[1.0, 0.75]$. MPQ($\lambda$) achieves performance better than MPPI ($H = 64$) with true dynamics and comparable to model-free RL in the limit for a wide range of bias factors $b$, and convergence rate is generally faster for smaller bias. For large values of $b$, MPQ($\lambda$) either fails to improve or

diverges as the compounding effects of model-bias hurt learning, making model-free RL the more favorable alternative. A similar trend is observed in Figures 3(a) and 3(b) where MPQ($\lambda$) outperforms MPPI with corresponding bias in the mass, inertia and friction coefficients of the pen with atleast a margin of over 30% in terms of success rate. It also achieves performance comparable to MPPI with true dynamics and model-free RL in the limit, but is unable to do so for $b = 1.0$. We conclude that while MPQ($\lambda$) is robust to large amount of model bias, if the model is extremely un-informative, relying on MPC can degrade performance.

**O 5.** MPQ($\lambda$) *is robust to planning horizon and number of trajectory samples in sampling-based MPC.* TD($\lambda$) based approaches are used for bias-variance trade-off for value function estimation in model-free RL. In our framework, $\lambda$ plays a similar role, but it trades-off bias due to the dynamics model and learned value function against variance due to long-horizon rollouts. We empirically quantify this on the CARTPOLESWINGUP task by training MPQ($\lambda$) with different values of horizon and number of particles for $\lambda = 1.0$ and $\lambda = 0.8$ respectively. Results in Fig. 2(g) show that - (1) using $\lambda$ can overcome effects of model-bias irrespective of the planning horizon (except for very small values of $H = 1$ or 2) and (2) using $\lambda$ can overcome variance due to limited number of particles with long horizon rollouts. The ablative study in Fig. 2(f) lends evidence to the fact that is preferable to simply decay $\lambda$ over time than trying to tune the discrete horizon value to balance model bias. Not only does decaying $\lambda$ achieve a better convergence rate and asymptotic performance than tuning horizon, the performance is more robust to different decay rates (as evidenced from Fig. 2(d)), whereas the same does not hold for varying horizon.

## 6 CONCLUSION

In this paper, we presented a general framework to mitigate model-bias in MPC by blending model-free value estimates using a parameter $\lambda$, to systematically trade-off different sources of error. Our practical algorithm achieves performance close to MPC with access to the *true* dynamics and asymptotic performance of model-free methods, while being sample efficient. An interesting avenue for future research is to vary $\lambda$ in a state-adaptive fashion. In particular, reasoning about the model and value function uncertainty may allow us to vary $\lambda$ to rely more or less on our model in certain parts of the state space. Another promising direction is to extend the framework to explicitly incorporate constraints by leveraging different constrained MPC formulations.

### ACKNOWLEDGMENTS

This work was supported in part by ARL SARA CRA W911NF-20-2-0095. The authors would like to thank Aravind Rajeswaran for help with code for the peg insertion task.

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

# A  APPENDIX

## A.1  PROOFS

We present upper-bounds on performance of a greedy policy when using approximate value functions and models. We also analyze the case of finite horizon planning with an approximate dynamics model and terminal value function which can be seen as a generalization of (Sun et al., 2018). For simplicity, we switch to using $\hat{V}(s)$ to the learnt model-free value function (instead of $\hat{Q}(s)$)

Let $\hat{V}(s)$ be an $\epsilon$-approximation $\left\|\hat{V}(s)-V_{\mathcal{M}}^{\pi^*}(s)\right\|_\infty \leq \epsilon$. Let MDP $\hat{\mathcal{M}}$ be an $\alpha$-approximation of $\mathcal{M}$ such that $\forall (s,a)$, we have $\left\|\hat{P}(s'|s,a)-P(s'|s,a)\right\|_1 \leq \alpha$ and $|\hat{c}(s,a)-c(s,a)| \leq \alpha$.

### A.1.1  A GENTLE START: BOUND ON PERFORMANCE OF 1-STEP GREEDY POLICY

**Theorem A.1.** *Let the one-step greedy policy be*

$$\hat{\pi}(s) = \operatorname*{argmin}_{a\in\mathcal{A}} \hat{c}(s,a) + \gamma\Sigma_{s'}\hat{P}(s'|s,a)\hat{V}(s') \tag{12}$$

*The performance loss of $\hat{\pi}(s)$ w.r.t optimal policy $\pi^*$ on MDP $\mathcal{M}$ is bounded by*

$$\left\|V_{\mathcal{M}}^{\hat{\pi}}(s)-V_{\mathcal{M}}^{\pi^*}(s)\right\|_\infty \leq \frac{2\left(\gamma\epsilon+\alpha+\gamma\alpha\left(\frac{V_{\max}-V_{\min}}{2}\right)\right)}{1-\gamma} \tag{13}$$

*Proof.* From (12) we have $\forall s \in \mathcal{S}$

$$\hat{c}(s,\hat{\pi}(s))+\gamma\sum_{s'}\hat{P}(s'|s,\hat{\pi}(s))\hat{V}(s') \leq \hat{c}(s,\pi^*(s))+\gamma\sum_{s'}\hat{P}(s'|s,\pi^*(s))\hat{V}(s')$$

$$\hat{c}(s,\hat{\pi}(s))-\hat{c}(s,\pi^*(s)) \leq \gamma\left(\sum_{s'}\hat{P}(s'|s,\pi^*(s))\hat{V}(s')-\sum_{s'}\hat{P}(s'|s,\hat{\pi}(s))\hat{V}(s')\right)$$

$$\left(\text{using } \left\|\hat{V}(s)-V_{\mathcal{M}}^{\pi^*}(s)\right\|_\infty \leq \epsilon\right)$$

$$\hat{c}(s,\hat{\pi}(s))-\hat{c}(s,\pi^*(s)) \leq \gamma\left(\sum_{s'}\hat{P}(s'|s,\pi^*(s))V_{\mathcal{M}}^{\pi^*}(s')-\sum_{s'}\hat{P}(s'|s,\hat{\pi}(s))V_{\mathcal{M}}^{\pi^*}(s')\right)+2\gamma\epsilon$$

$$\left(\text{using } |\hat{c}(s,a)-c(s,a)| \leq \alpha\right)$$

$$c(s,\hat{\pi}(s))-c(s,\pi^*(s)) \leq 2\gamma\epsilon+2\alpha+\gamma\left(\sum_{s'}\hat{P}(s'|s,\pi^*(s))V_{\mathcal{M}}^{\pi^*}(s')-\sum_{s'}\hat{P}(s'|s,\hat{\pi}(s))V_{\mathcal{M}}^{\pi^*}(s')\right)$$
$$\tag{14}$$

Now, let $s$ be the state with the max loss $V_{\mathcal{M}}^{\hat{\pi}}(s)-V_{\mathcal{M}}^{\pi^*}(s)$,

$$V_{\mathcal{M}}^{\hat{\pi}}(s)-V_{\mathcal{M}}^{\pi^*}(s) = c(s,\hat{\pi})-c(s,\pi^*)+\gamma\sum_{s'}\left(P(s'|s,\hat{\pi})V_{\mathcal{M}}^{\hat{\pi}}(s')-P(s'|s,\pi^*)V_{\mathcal{M}}^{\pi^*}(s')\right)$$

Substituting from (14)

$$V_{\mathcal{M}}^{\pi^*}(s) - V_{\mathcal{M}}^{\hat{\pi}}(s) \leq 2\gamma\epsilon + 2\alpha + \gamma \sum_{s'} \hat{P}(s'|s,\pi^*(s)) V_{\mathcal{M}}^{\pi^*}(s') - \gamma \sum_{s'} \hat{P}(s'|s,\hat{\pi}(s)) V_{\mathcal{M}}^{\pi^*}(s')$$
$$- \gamma \sum_{s'} P(s'|s,\pi^*) V_{\mathcal{M}}^{\pi^*}(s') + \gamma \sum_{s'} P(s'|s,\hat{\pi}) V_{\mathcal{M}}^{\hat{\pi}}(s')$$

Add and subtract $\gamma \sum_{s'} P(s'|s,\hat{\pi}) V_{\mathcal{M}}^{\pi^*}(s')$ and re-arrange

$$V_{\mathcal{M}}^{\pi^*}(s) - V_{\mathcal{M}}^{\hat{\pi}}(s) \leq 2\gamma\epsilon + 2\alpha + \gamma \sum_{s'} \left( \hat{P}(s'|s,\pi^*) - P(s'|s,\pi^*) \right) V_{\mathcal{M}}^{\pi^*}(s')$$
$$- \gamma \sum_{s'} \left( \hat{P}(s'|s,\hat{\pi}) - P(s'|s,\hat{\pi}) \right) V_{\mathcal{M}}^{\pi^*}(s')$$
$$+ \gamma \sum_{s'} P(s'|s,\hat{\pi}) \left( V_{\mathcal{M}}^{\hat{\pi}}(s') - V_{\mathcal{M}}^{\pi^*}(s') \right)$$
$$\leq 2\gamma\epsilon + 2\alpha + 2\gamma\alpha \left( \frac{V_{\max} - V_{\min}}{2} \right) + \gamma \sum_{s'} P(s'|s,\hat{\pi}) \left( V_{\mathcal{M}}^{\hat{\pi}}(s') - V_{\mathcal{M}}^{\pi^*}(s') \right)$$

Since $s$ is the state with largest loss

$$\left\| V_{\mathcal{M}}^{\pi^*}(s) - V_{\mathcal{M}}^{\hat{\pi}}(s) \right\|_\infty \leq 2\gamma\epsilon + 2\alpha + 2\gamma\alpha \left( \frac{V_{\max} - V_{\min}}{2} \right) + \gamma \sum_{s'} P(s'|s,\hat{\pi}) \left\| V_{\mathcal{M}}^{\pi^*}(s) - V_{\mathcal{M}}^{\hat{\pi}}(s) \right\|_\infty$$
$$\leq 2\gamma\epsilon + 2\alpha + 2\gamma\alpha \left( \frac{V_{\max} - V_{\min}}{2} \right) + \gamma \left\| V_{\mathcal{M}}^{\pi^*}(s) - V_{\mathcal{M}}^{\hat{\pi}}(s) \right\|_\infty$$

Re-arranging terms we get

$$\left\| V_{\mathcal{M}}^{\pi^*}(s) - V_{\mathcal{M}}^{\hat{\pi}}(s) \right\|_\infty \leq \frac{2 \left( \gamma\epsilon + \alpha + \gamma\alpha \left( \frac{V_{\max} - V_{\min}}{2} \right) \right)}{1 - \gamma} \tag{15}$$

which concludes the proof. $\qquad\square$

### A.1.2 BOUND ON PERFORMANCE OF H-STEP GREEDY POLICY

**Notation:** For brevity let us define the following macro,

$$\langle V, \pi, \mathcal{M} \rangle_H = \mathbb{E}_{\mu_{\mathcal{M}}^\pi} \left[ \sum_{i=0}^{H-1} \gamma^i c(s_i, a_i) + \gamma^H V(s_H) \right] \tag{16}$$

which represents the expected cost achieved when executing policy $\pi$ on $\mathcal{M}$ using $V$ as the terminal cost. We can substitute different policies, terminal costs and MDPs. For example, $\left\langle \hat{V}, \hat{\pi}, \hat{\mathcal{M}} \right\rangle_H$ is the expected cost obtained by running policy $\hat{\pi}$ on simulator $\hat{\mathcal{M}}$ for $H$ steps with approximate learned terminal value function $\hat{V}$.

**Lemma A.1.** *For a given policy $\pi$, the optimal value function $V_{\mathcal{M}}^{\pi^*}$ and MDPs $\mathcal{M}, \hat{\mathcal{M}}$ the following performance difference holds*

$$\left\| \left\langle V_{\mathcal{M}}^{\pi^*}, \pi, \mathcal{M} \right\rangle_H - \left\langle V_{\mathcal{M}}^{\pi^*}, \pi, \hat{\mathcal{M}} \right\rangle_H \right\|_\infty \leq \gamma \left( \frac{1 - \gamma^{H-1}}{1 - \gamma} \right) \alpha H \left( \frac{c_{\max} - c_{\min}}{2} \right) + \gamma^H \alpha H \left( \frac{V_{\max} - V_{\min}}{2} \right) + \frac{1 - \gamma^H}{1 - \gamma} \alpha$$

*Proof.* We temporarily introduce a new MDP $\mathcal{M}'$ that has the same cost function as a $\mathcal{M}$, but transition function of $\hat{\mathcal{M}}$

$$\left\langle V_{\mathcal{M}}^{\pi^*}, \pi, \mathcal{M} \right\rangle_H - \left\langle V_{\mathcal{M}}^{\pi^*}, \pi, \hat{\mathcal{M}} \right\rangle_H = \left\langle V_{\mathcal{M}}^{\pi^*}, \pi, \mathcal{M} \right\rangle_H - \left\langle V_{\mathcal{M}}^{\pi^*}, \pi, \mathcal{M}' \right\rangle_H$$
$$+ \left\langle V_{\mathcal{M}}^{\pi^*}, \pi, \mathcal{M}' \right\rangle_H - \left\langle V_{\mathcal{M}}^{\pi^*}, \pi, \hat{\mathcal{M}} \right\rangle_H \tag{17}$$

Let $\Delta P(s_0...s_H) = P(s_0...s_H) - \hat{P}(s_0...s_H)$ represent the difference in distribution of states encountered by executing $\pi$ on $\mathcal{M}$ and $\hat{\mathcal{M}}$ respectively starting from state $s_0$.

Expanding the RHS of (17)

$$= \sum_{s_0,...,s_H} \Delta P(s_0...s_H) \left( \sum_{i=0}^{H-1} \gamma^i c(s_i,a_i) + \gamma^H V_{\mathcal{M}}^{\pi^*}(s_H) \right) + \mathbb{E}_{\mu_{\hat{\mathcal{M}}}^{\pi}} \left[ \sum_{i=0}^{H-1} \gamma^i (c(s_i,a_i) - \hat{c}(s_i,a_i)) \right] \quad (18)$$

Since the first state $s_1$ is the same

$$= \sum_{s_1,...,s_H} \Delta P(s_1...s_H) \left( \sum_{i=1}^{H-1} \gamma^i c(s_i,a_i) + \gamma^H V_{\mathcal{M}}^{\pi^*}(s_H) \right) + \mathbb{E}_{\mu_{\hat{\mathcal{M}}}^{\pi}} \left[ \sum_{i=0}^{H-1} \gamma^i (c(s_i,a_i) - \hat{c}(s_i,a_i)) \right]$$

$$\leq \left\| \sum_{s_1,...,s_H} \Delta P(s_1...s_H) \left( \sum_{i=1}^{H-1} \gamma^i c(s_i,a_i) + \gamma^H V_{\mathcal{M}}^{\pi^*}(s_H) \right) \right\|_{\infty} + \left\| \mathbb{E}_{\mu_{\hat{\mathcal{M}}}^{\pi}} \left[ \sum_{i=0}^{H-1} \gamma^i (c(s_i,a_i) - \hat{c}(s_i,a_i)) \right] \right\|_{\infty}$$

$$\leq \left\| \sum_{s_1,...,s_H} \Delta P(s_1...s_H) \left( \sum_{i=1}^{H-1} \gamma^i c(s_i,a_i) + \gamma^H V_{\mathcal{M}}^{\pi^*}(s_H) \right) \right\|_{\infty} + \frac{1-\gamma^H}{1-\gamma} \alpha$$

$$(19)$$

where the first inequality is obtained by applying the triangle inequality and the second one is obtained by applying triangle inequality followed by the upper bound on the error in cost-function.

$$\leq \left\| \sum_{s_2,...,s_{H+1}} \Delta P(s_2...s_{H+1}) \right\|_{\infty} \sup \left( \sum_{i=1}^{H-1} \gamma^i c(s_i,a_i) + \gamma^H V_{\mathcal{M}}^{\pi^*}(s_H) - K \right) + \frac{1-\gamma^H}{1-\gamma} \alpha \quad (20)$$

By choosing $K = \sum_{i=1}^{H-1} \gamma^i (c_{\max} + c_{\min})/2 + \gamma^H (V_{\max} + V_{\min})/2$ we can ensure that the term inside sup is upper-bounded by $\gamma(1-\gamma^{H-1})/(1-\gamma)((c_{\max}-c_{\min})/2) + \gamma^H (V_{\max}-V_{\min})/2$

$$\leq \gamma \left( \frac{1-\gamma^{H-1}}{1-\gamma} \right) \alpha H \left( \frac{c_{\max}-c_{\min}}{2} \right) + \gamma^H \alpha H \left( \frac{V_{\max}-V_{\min}}{2} \right) + \frac{1-\gamma^H}{1-\gamma} \alpha \quad (21)$$

$$\square$$

The above lemma builds on similar results in (Kakade et al., 2003; Abbeel & Ng, 2005; Ross & Bagnell, 2012).

We are now ready to prove our main theorem, i.e. the performance bound of an MPC policy that uses an approximate model and approximate value function.

**Proof of Theorem 3.1**

*Proof.* Since, $\hat{\pi}$ is the greedy policy when using $\hat{\mathcal{M}}$ and $\hat{V}$,

$$\begin{aligned} \left\langle \hat{V}, \hat{\pi}, \hat{\mathcal{M}} \right\rangle_H &\leq \left\langle \hat{V}, \pi^*, \hat{\mathcal{M}} \right\rangle_H \\ \left\langle V_{\mathcal{M}}^{\pi^*}, \hat{\pi}, \hat{\mathcal{M}} \right\rangle_H &\leq \left\langle V_{\mathcal{M}}^{\pi^*}, \pi^*, \hat{\mathcal{M}} \right\rangle_H + 2\gamma^H \epsilon \text{ (using } \left\| \hat{V} - V_{\mathcal{M}}^{\pi^*} \right\|_1 \leq \epsilon ) \end{aligned} \quad (22)$$

Also, we have

$$\begin{aligned} \left\langle V_{\mathcal{M}}^{\pi^*}, \hat{\pi}, \mathcal{M} \right\rangle_H - \left\langle V_{\mathcal{M}}^{\pi^*}, \pi^*, \mathcal{M} \right\rangle_H &= \left( \left\langle V_{\mathcal{M}}^{\pi^*}, \hat{\pi}, \mathcal{M} \right\rangle_H - \left\langle V_{\mathcal{M}}^{\pi^*}, \hat{\pi}, \hat{\mathcal{M}} \right\rangle_H \right) \\ &\quad - \left( \left\langle V_{\mathcal{M}}^{\pi^*}, \pi^*, \mathcal{M} \right\rangle_H - \left\langle V_{\mathcal{M}}^{\pi^*}, \pi^*, \hat{\mathcal{M}} \right\rangle_H \right) \\ &\quad + \left( \left\langle V_{\mathcal{M}}^{\pi^*}, \hat{\pi}, \hat{\mathcal{M}} \right\rangle_H - \left\langle V_{\mathcal{M}}^{\pi^*}, \pi^*, \hat{\mathcal{M}} \right\rangle_H \right) \end{aligned} \quad (23)$$

The first two terms can be bounded using Lemma A.1 and the third term using Eq. (22) to get

$$\left\langle V_{\mathcal{M}}^{\pi^*}, \hat{\pi}, \mathcal{M} \right\rangle_H - \left\langle V_{\mathcal{M}}^{\pi^*}, \pi^*, \mathcal{M} \right\rangle_H$$
$$\leq 2\left( \gamma \frac{1-\gamma^{H-1}}{1-\gamma} \alpha H \left( \frac{c_{\max} - c_{\min}}{2} \right) + \gamma^H \alpha H \left( \frac{V_{\max} - V_{\min}}{2} \right) + \frac{1-\gamma^H}{1-\gamma} \alpha + \gamma^H \epsilon \right) \tag{24}$$

Now, let $s$ be the state with max loss $V_{\mathcal{M}}^{\hat{\pi}}(s) - V_{\mathcal{M}}^{\pi^*}(s)$

$$V_{\mathcal{M}}^{\hat{\pi}}(s) - V_{\mathcal{M}}^{\pi^*}(s) = \left\langle V_{\mathcal{M}}^{\hat{\pi}}, \hat{\pi}, \mathcal{M} \right\rangle_H - \left\langle V_{\mathcal{M}}^{\pi^*}, \pi^*, \mathcal{M} \right\rangle_H$$
$$= \left( \left\langle V_{\mathcal{M}}^{\hat{\pi}}, \hat{\pi}, \mathcal{M} \right\rangle_H - \left\langle V_{\mathcal{M}}^{\pi^*}, \hat{\pi}, \mathcal{M} \right\rangle_H \right) + \left( \left\langle V_{\mathcal{M}}^{\pi^*}, \hat{\pi}, \mathcal{M} \right\rangle_H - \left\langle V_{\mathcal{M}}^{\pi^*}, \pi^*, \mathcal{M} \right\rangle_H \right)$$
$$= \gamma^H \left( V_{\mathcal{M}}^{\hat{\pi}}(s_{H+1}) - V_{\mathcal{M}}^{\pi^*}(s_{H+1}) \right) + \left( \left\langle V_{\mathcal{M}}^{\pi^*}, \hat{\pi}, \mathcal{M} \right\rangle_H - \left\langle V_{\mathcal{M}}^{\pi^*}, \pi^*, \mathcal{M} \right\rangle_H \right)$$
$$\leq \gamma^H \left( V_{\mathcal{M}}^{\hat{\pi}}(s) - V_{\mathcal{M}}^{\pi^*}(s) \right)$$
$$+ 2\left( \frac{\gamma(1-\gamma^{H-1})}{1-\gamma} \alpha H \left( \frac{c_{\max} - c_{\min}}{2} \right) + \gamma^H \alpha H \left( \frac{V_{\max} - V_{\min}}{2} \right) + \frac{1-\gamma^H}{1-\gamma} \alpha + \gamma^H \epsilon \right)$$
$$\tag{25}$$

where last inequality comes from applying Eq. (24) and the fact that $s$ is the state with max loss. The final expression follows from simple algebraic manipulation. $\square$

## A.2 EXPERIMENT DETAILS

### A.2.1 TASK DETAILS

**CARTPOLESWINGUP**

- Reward function: $x_{cart}^2 + \theta_{pole}^2 + 0.01 v_{cart} + 0.01 \omega_{pole} + 0.01 a^2$

- Observation: $[x_{cart}, \theta_{pole}, v_{cart}, \omega_{pole}]$ (4 dim)

**SAWYERPEGINSERTION** We simulate sensor noise by placing a simulated position sensor at the target location in the MuJoCo physics engine that adds Gaussian noise with $\sigma = 10$cm to the observed 3D position vector. MPPI uses a deterministic model that does not take sensor noise into account for planning. Every episode lasts for 75 steps with a timestep of 0.02 seconds between steps

- Reward function: $-1.0 * ||x_{ee} - x_{target}||_1 - 5.0 * ||x_{ee} - x_{target}||_2 + 5 * \mathbb{1}(||x_{ee} - x_{target}||_2 < 0.06)$

- Observation: $\left[ q_{pos}, q_{vel}, x_{ee}, x_{target}, x_{ee} - x_{target}, ||x_{ee} - x_{target}||_1, ||x_{ee} - x_{target}||_2 \right]$ (25 dim)

An episode is considered successful if the peg stays within the hole for atleast 5 steps.

**INHANDMANIPULATION** This environment was used without modification from the accompanying codebase for Rajeswaran* et al. (2018) and is available at https://bit.ly/3f6MNBP

- Reward function: $-||x_{obj} - x_{des}||_2 + z_{obj}^T z_{des} +$ Bonus for proximity to desired pos + orien, $z_{obj}^T z_{des}$ represents dot product between object axis and target axis to measure orientation similarity.

- Observation: $[q_{pos}, x_{obj}, v_{obj}, z_{obj}, z_{des}, x_{obj} - x_{des}, z_{obj} - z_{des}]$ (45 dim)

Every episode lasts 75 steps with a timestep of 0.01 seconds between steps. An episode is considered successful if the orientation of the pen stays within a specified range of desired orientation for atleast 20 steps. The orientation similarity is measured by the dot product between the pen's current longitudinal axis and desired with a threshold of 0.95.

Table 1: MPPI Parameters

| CARTPOLESWINGUP | | SAWYERPEGINSERTION | | INHANDMANIPULATION | |
|---|---|---|---|---|---|
| **Parameter** | **Value** | **Parameter** | **Value** | **Parameter** | **Value** |
| Horizon | 32 | Horizon | 20 | Horizon | 32 |
| Num particles | 60 | Num particles | 100 | Num particles | 100 |
| Covariance ($\Sigma$) | 0.45 | Covariance ($\Sigma$) | 0.25 | Covariance ($\Sigma$) | 0.3 |
| Temperature($\beta$) | 0.1 | Temperature($\beta$) | 0.1 | Temperature($\beta$) | 0.15 |
| Filter coeffs | [1.0, 0.0, 0.0] | Filter coeffs | [0.25, 0.8, 0.0] | Filter coeffs | [0.25, 0.8, 0.0] |
| Step size | 1.0 | Step size | 0.9 | Step size | 1.0 |
| $\gamma$ | 0.99 | $\gamma$ | 0.99 | $\gamma$ | 0.99 |

### A.2.2 LEARNING DETAILS

**Validation**: Validation is performed after every $N$ training episodes during training for $N_{\text{eval}}$ episodes using a fixed set of start states that the environment is reset to. We ensure that the same start states are sampled at every validation iteration by setting the seed value to a pre-defined validation seed, which is kept constant across different runs of the algorithm with different training seeds. This helps ensure consistency in evaluating different runs of the algorithm. For all our experiments we set $N = 40$ and $N_{\text{eval}} = 30$.

**MPQ**($\lambda$): For all tasks, we represent Q function using 2 layered fully-connected neural network with 100 units in each layer and ReLU activation. We use ADAM (Kingma & Ba, 2014) for optimization with a learning rate of 0.001 and discount factor $\gamma = 0.99$. Further, the buffer size is 1500 for CARTPOLESWINGUP and 3000 for the others, with batch size of 64 for all. We smoothly decay $\lambda$ according to the following sublinear decay rate

$$\lambda_t = \frac{\lambda_0}{1 + \kappa\sqrt{t}} \tag{26}$$

where the decay rate $\kappa$ is calculate based on the desired final value of $\lambda$. For batch size we did a search from [16, 64] with a step size of 16 and buffer size was chosen from 1500, 3000, 5000. While batch size was tuned for cartpole and then fixed for the remaining two environments, the buffer size was chosen independently for all three.

**Proximal Policy Optimization (PPO)**: Both policy and value functions are represented by feed forward networks with 2 layers each with 64 and 128 units for policy and value respectively. All other parameters are left to the default values. The number of trajectories collected per iteration is modified to correspond with the same number of samples collected between validation iterations for MPQ($\lambda$). We collect 40 trajectories per iteration. Asymptotic performance is reported as average of last 10 validation iterations after 500 training iters in SAWYERPEGINSERTION and 2k in INHANDMANIPULATION.

**MPPI parameters** Table 1 shows the MPPI parameters used for different experiments. In addition to the standard MPPI parameters, in certain cases we also use a step size parameter as introduced by Wagener et al. (2019). For INHANDMANIPULATION and SAWYERPEGINSERTION we also apply autoregressive filtering on the sampled MPPI trajectories to induce smoothness in the sampled actions, with tuned filter coefficients. This has been found to be useful in prior works (Summers et al., 2020; Lowrey et al., 2018) for getting MPQ($\lambda$) to work on high dimensional control tasks. The temperature, initial covariance and step size parameters for MPPI were tuned using a grid search with true dynamics. Temperature and initial covariance were set within the range of [0.0,1.0] and step size from [0.5,1.0] with a discretization of 0.05. The number of particles were searched from [40,120] with a step size of 10 and the horizon was chosen from 4 different values [16,20,32,64]. The best performing parameters then chosen based on average reward over 30 episodes with a fixed seed value to ensure reproducibility. The same parameters were then used in the case of biased dynamics and MPQ($\lambda$), to clearly demonstrate that MPQ($\lambda$) can overcome sources of error in the base MPC implementation.

