# OpenReview forum: "Blending MPC & Value Function Approximation for Efficient Reinforcement Learning"
_ICLR.cc/2021/Conference — ICLR 2021 Poster_

### Official Review · AnonReviewer1 · 2020-10-19
**Neat idea and clearly written paper but evaluations can be improved**

**Rating:** 6
**Confidence:** 4

**Review:**

##
**Summary:**
The paper proposes to combine MPC and model-free RL to overcome the possible modelling errors. Thereby the approach achieves the sample-efficiency of MPC and the control quality of model-free RL. The resulting MPQ(\lambda) algorithm uses MPPI to obtain the actions by optimizing the blended MPC objective. The Q-targets for fitting the q function also use the blended Q-estimate.


##
**Quality, Originality & Significance:**
The idea of combining MPC and model-free RL is straight forward and not novel (the paper also does not claim this). However, the exact instantiation is novel, very well motivated and feels natural. My biggest concerns are the experiments. The cartpole experiments show the improved performance compared to MPPI on the biased model and the impact of the lambda and model bias. However, the PPO baseline is missing for the cartpole, right? Furthermore, is PPO a fair comparison for the MPQ($\lambda$) algorithm to evaluate sample complexity? While PPO is a batched update, the MPQ($\lambda$) uses step based updates. Wouldn't a model-free step-based update algorithm such as DDPG, SAC etc. be a better baseline to evaluate the improved sample complexity? Regarding the high-dimensional tasks, the provided evaluations do not enable an evaluation whether the task is solved or not. Could you please provide videos of the final policies, otherwise the achieved reward is just a random number. Furthermore, the paper shows confidence bounds for the MPPI baselines but not for the MPQ($\lambda$) algorithm. Also the learning curves are cut before converging, could you train every instance until convergence? Furthermore, could you please include the asymptotic performance of your baseline in the plots. The definition of 'validation iteration' remains unclear. Given the current evaluations the stated claim of applicability to high-dim tasks cannot be made as the evaluations are not sufficient. The used modelling bias is also very limited as the paper only compares to biases of the model parameters but not against other sources of biases. Ultimately the increased performance can only be shown on the physical system.

##
**Clarity & Style:**
The paper is really well written and understandable! A few sections could be improved, e.g., text between Eq. 6 & Eq. 8. In this section it is a bit unclear what is expanded and how it is expanded. It would be beneficial to rethink the labeling of the Q-functions as they can be quite confusing. Maybe a table of the different subscript/superscript definitions would simplify the reading, as I had to search for the exact definitions frequently. Furthermore, there are minor styling issues:
* Inline equations are consuming too much space to mess up line spacing, e.g, Section 2.1 argmin, Theorem 3.1, norms,
* the min in Equation 13 needs two spaces and a subscript
* Experiment O4 ends with two dots
* White space around figure 2 can be optimized

##
**Conclusion:**
All in all the paper is nicely written with a clear and well motivated idea of combining MPC & model-free rl. Right now the main problem is the execution of the evaluations. The performance on the high-dim tasks is unclear and the baseline is missing for the cartpole. I would be happy to improve my score to
* **weak accept** if a step based model free RL algorithm is added to cartpole, claims regarding high-dim tasks are adapted and videos of the high-dimensional tasks are released.
* **accept** if the high-dimensional tasks are working properly with MPQ($\lambda$)
* **strong accept** if MPQ($\lambda$) shows this performance on a physical system


P.S. You might also try to get medium dimension tasks working such as hopper or cheetah. That might be a bit easier.

##
**Post Discussion Comments:** So the author did a **filibuster** and **flooded the discussions** with bloated comments. In this manner it was close to impossible to keep track of anything. **There has to be character limit for responses otherwise this is not feasible**. I looked at the videos and your physics simulators looks **really catchy**. At one point in time, the pole of the cartpole is at 10-11 o'clock and the cart starts moving right (the pole has close to no velocity and hence only a very small angular momentum). In this setting it would be natural that the pole would fall down if this state is maintained for a longer period (which it is in the video). However the pole goes upwards into the balancing position. This is really weird. And don't get me started on the pen-orientation as the pen sometimes floats mid-air. For this setting the gravitational constant really does not seem right. I also wouldn't consider the task solved as this is more an really uncoordinated movements for three specific configurations.

For the simulation studies some doubts remain, but the authors improved the paper. Therefore, I am going to increase my score to weak accept. Nevertheless, the experimental evaluation could be improved and the paper would really benefit from real experiments.

---

> ### Author Response · Authors · 2020-11-21
> **Response to review**
>
> ### Experimental Evaluation
> ***************************
> 1. **Model-free baselines, task success evaluation and updated learning curves**
>
>      - We have updated the paper draft to include the **SAC baseline** for all three environments. Please refer to the “Overall Response” comment for more details.
>
>      - MPQ($\lambda$) and model-free baselines were **re-trained for a larger number of iterations** and the curves have been updated accordingly.
>
>      - **Asymptotic performance** of model-free baseline has also been included (defined as average reward/success of last 10 iterations).
>
>      - **Videos of final policies** after MPQ($\lambda$) training have been uploaded in the supplementary material.
>
>      - To further demonstrate the ability of MPQ($\lambda$) in solving the high dimensional problems, we have also provided **curves for average success rate** achieved by different algorithms on the InHandManipulation and SawyerPegInsertion. Please refer to the “Overall Response” comment for a detailed discussion. The updated results further solidify our claim that MPQ($\lambda$) can mitigate large amounts of model-bias in MPC even in high-dimensional problems.
>
>      - **Error bars in terms of standard error of the mean** have also been provided for all algorithms. We chose this metric to denote confidence over the estimated average performance of the algorithm by normalizing for finite samples. We believe this is a more representative metric than standard deviation, since it quantifies a bound on average case performance of our algorithm.
>
>
> With these updates to the draft we believe we have fulfilled the reviewer’s criteria for an **accept**. We would be happy to answer any follow-up queries about the experimental results. Next, we address the remaining remarks from the reviewer.
>
>
> 2. **Meaning of validation iteration**
>
>     Validation is performed after every $N$ training episodes during training for $N_{eval}$ episodes using a fixed set of start states that the environment is reset to. We ensure that the same start states are sampled at every validation iteration by setting the seed value to a pre-defined validation seed, which is kept constant across different runs of the algorithm with different training seeds. This helps ensure consistency in evaluating different runs of the algorithm. In all our experiments we use $N=40$ and $N_{eval}=30$. We have added this discussion to the appendix.
>
>
> 3. **Modelling Bias**
>
>     We agree that the true test for the efficacy of MPQ($\lambda$) is deployment on a physical system, which is our top priority for future work. Although, model bias can manifest in MPC algorithms in several different ways, through our experiments we have tried to test MPQ($\lambda$) on qualitatively different forms of biased models that often occur in the real-world, with the common thread being that biased models can lead to *persistent errors*. In the CartpoleSwingup and InhandManipulation tasks, model bias occurs in the form of unknown dynamics parameters. This is a common problem in robotic control tasks, especially ones that involve a robot manipulating an unknown object. In CartpoleSwingup we specifically set the masses of the cart and pole to lower values than reality which biases our controller to *persistently input smaller controls* than required to swingup the pendulum. On the contrary, in InhandManipulation, we set the mass, inertia and friction coefficients of the pen to be higher than true values. This causes the MPC to *persistently optimize for overly aggressive policies* making it hard to recover from mistakes. The PegInsertion task tests the ability of MPQ($\lambda$) to adapt to errors in perception. Here, we simulate a noisy sensor at the target location inside the hole which provides position measurements corrupted by Gaussian noise with a large standard deviation of 10cm. The MPC algorithm does not simulate sensor noise and uses the current measurement as the true target. This causes MPC to *persistently miss the target* and hover around the hole location. Through our experiments, we demonstrate that MPQ($\lambda$) can successfully overcome qualitatively different kinds of model bias by blending in a value function learned from real-world data.
>
> ### Clarity & Style
> *****************
> We would like to thank the reviewer for the helpful comments on improving clarity, style and other grammatical errors. We will try to incorporate as many of the suggestions as possible in the final draft given the space constraints.
>
> We hope that we have sufficiently addressed the reviewer’s concerns and incorporated all the suggested updates to experimental evaluation. We would be happy to answer any remaining queries that the reviewer might have.

---

> ### Comment · AnonReviewer4 · 2020-11-25
> **Reviewer response**
>
> +1 for character limit. I am trying to keep up with all the discussions, but there is a lot here when reviewing multiple papers and having a life. I think the authors did it with good intents, but it is hard.
>
> I agree with this reviewers conclusions.

---

### Official Review · AnonReviewer4 · 2020-10-23
**Interesting paper blending MPC with RL, limited by experimental evaluation.**

**Rating:** 6
**Confidence:** 3

**Review:**

Summary:

The paper provides and interesting analysis of and new method for model-predictive control in reinforcement learning.
Specifically, it proposes a new framework MPQ(lambda) for joining model-based MPC with learned value estimates in RL.
The authors develop an formulation to find an optimal prediction horizon and and how this works in an online reinforcement learning framework.
The new approach is evaluated on 3 continuous control tasks and compared to some other baselines.

-----
Score reasoning:

This paper has interesting theoretical contributions to multiple areas of machine learning: model-based RL, MPC, and value estimation, but the somewhat limited experimental evaluation make the efficacy of the method more difficult to judge. Overall, the paper is well written and I enjoyed reading it. I now will address my conceptual comments followed by more minor suggestions.

-----
Rebuttal update: the authors have gone beyond the normal scope of a rebuttal phase to update their experiments and the motivation of the work, and for that reason I have improved my recommendation to be above the acceptance threshold.

-----
Experimental Validation Questions.
I am breaking this section of the review into it's own section because it is where the majority of my questions are.

E1) The authors bring up model-based RL algorithms, but do not baseline against other algorithms generally considered sample efficient. PPO is not always easy to use (authors mention it not converging, and not substantial parameter tuning), how about SAC?
E1b) It would be very interesting to compare something similar to the PETS optimizer for MPC (cited in intro, but not really mentioned). These baseline changes could make the results much more believable.

E2) "All parameters were found using a coarse grid search" This makes the results suspect to me. Please clarify how coarse? Is the same search space used for all algorithms? Were defaults used for algorithms with previously published results? Do the results match?

E3) Was tuning of the reward functions done by hand in A.2.1, or are they referenced elsewhere? Are states like x-position and pole angle normalized in cartpole? This can have bigger effects in more complicated environments.

E4) "shaded regions represent standard deviation" for MPPI, is this over the same 30x3 evaluations? Very important to standard dev. is the number of samples.

E5) Cartpole swing up is a very similar task. It seems to be the only one where MPQ(lambda) substantially outperforms the baselines given (no error bars on MPQ too). How do the ablation studies of figure 2 reproduce on more challenging environments?

-----
Comments:
1) The authors refer to MPC as a "simpler, more practical alternative" to RL or a "more pragmatic approach" for "simple" policies.
Some would argue that RL is a simpler approach because it does not require any model in the case of model-free RL.
MPC also has many design decisions such as which optimizer to use or the planning horizon (multiple papers written on this topic).
I would like the authors to explain this with more detail, or defend their stance.

2) The authors may consider including these two other papers that relate to model-based RL, model-bias, and MPC horizon https://arxiv.org/pdf/2009.09593.pdf, https://arxiv.org/pdf/2002.04523.pdf.

3) How does model bias differ from model inaccuracy?
In MBRL, model-bias often refers to the model being more accurate in some areas of the state-space than others, and how this impacts the downstream ranking of action choices.
Do the authors consider this difference at all? How does model-accuracy drop when the bias terms are introduced in some basic metrics like mean-squared-error or negative-log-likelihood (metrics used in MBRL to quantify model-accuracy).

4) The position of the contribution in related works could be made stronger. I was unaware that MPQ was not the proposal of this paper until section 4. The difference between the two and why this matters should be in the introduction (unless the authors decide to add a dedicated related works section).
4b) how does entropy-regularized formulation impact the results? From my reading, that is an important part of the original MPQ paper, so I think it should be explained.

5) The conclusion to this paper is weak. It re-iterates what is done, but the authors should make a case how this impacts developments in robotics & control to better match up with the experiments and introduction. What should I take away from studying this paper?

6) It would be interesting to see the authors propose how to combine the MPQ framework with other forms of MPC that don't have an implicit terminal cost included. This may be for future work, but I would be interested in a comment.

-----
Minor comments:
1) There are some typos that impede reading, but overall the paper is well written.
- intro, paragraph 2, "owing to its ability to" is weird
- section 2.2 "since it plans..." it is vague here
- some missing commas in first paragraph of section 3 "First..."
- Missing period at end of paragraph "Baselines", missing period Figure 3 end, double period before Conclusion
- Typo in PPO A.2.2 "The"

2) in 3.2, the authors show how to blend the model-based and model-free methods, but point to a reference that is not obviously connected to me and call the approach "common".
I would suggest adding more references, or adjusting the claim.

3) Why was MPPI chosen as the MPC algorithm? It is a suitable choice, but could be added.

4) there is a lot of visuals in Figures 2 and 3. Maybe have fewer lines? The font could be enlarged and it is very confusing that the y-axis's are not all the same for similar data types.
4b) Figure 4b) has strange shading from the MPPI variance - it's not readable.

---

> ### Author Response · Authors · 2020-11-21
> **Response to review (1/4)**
>
> We would like to thank the reviewer for providing valuable feedback on our paper. We address your concerns below
>
> ### Experimental Evaluation
> ***************************
>
> E1)
>  - **Model-free Baselines:**
> We have updated the paper draft to include the SAC baseline for all three environments and the new results further strengthen our claim regarding the sample efficiency of MPQ($\lambda$) versus model-free RL. Please refer to the “Overall Response” comment for more details.
>
>
>  - **Comparison to PETS Optimizer**
> We stress that the main motivation of our work is to improve upon MPC from experience in order to mitigate errors introduced due to model bias, value function error and limited horizon. Our proposed MPQ($\lambda$) framework is theoretically well-founded and systematically trades-off different sources of errors in MPC leading to improved performance over time. In this work, our aim was not to benchmark different MPC algorithms against each other. We provide a novel interpretation of MPC as Q-function approximation and a framework to combine it with learned value functions that can be used with any underlying MPC algorithm. While it is true that a better MPC optimizer could potentially improve performance and is a great avenue for future work, an exhaustive benchmarking of different MPC algorithms is beyond the scope of this paper. Please refer to “Overall Response” for a detailed discussion.
>
> E2) **Hyperparameter Tuning Details:**
>
>  - MPPI: The temperature, initial covariance and step size parameters for MPPI were tuned with true dynamics using a coarse grid search. Temperature and initial covariance were searched within the range of [0.0, 1.0] and step size from [0.5,1.0] with a discretization of 0.05 for all of them. The number of particles were searched from [40,120] with a step size of 10 and the horizon was chosen from 4 different values from [16, 20, 32, 64]. The best performing parameters were then chosen based on average reward over 30 episodes with a fixed seed value to ensure reproducibility. The same parameters were used in the case of biased dynamics and MPQ(\lambda), to clearly demonstrate that MPQ($\lambda$) can overcome sources of error in the base MPC implementation.
>
>  - While the CartpoleSwingup and PegInsertion environments do not have previously published results, the InhandManipulation environment was first introduced in [1] and was used without any modification. MPPI has previously been used to solve this environment in [5], and our results match their reported performance in terms of average reward and success rate as shown in Figure 3(a) and 3(b). Furthermore, we have provided curves for average success rate for PegInsertion (Figure 3(d)) along with asymptotic performance of model-free baselines to show that our implementation achieves strong performance on all the tasks.
>
>  - The only parameters in MPQ($\lambda$) that were tuned were batch size, buffer size and $\lambda$ decay rate. For batch size we did a search from [16, 64] with a step size of 16 and buffer size was chosen from {1500, 3000, 5000}.  While batch size was tuned for cartpole and then fixed for the remaining two environments, the buffer size was chosen independently for all three.
>
> We have added these details to the appendix of the revised draft. Further, we would like to note that even though better settings of parameters might exist, the current settings are sufficient to prove the efficacy of MPQ($\lambda$) in overcoming the major sources of bias (from model and terminal cost) in MPC. This can be evidenced from the following
>  - In CartpoleSwingup, MPPI with true dynamics and $H=32$ achieves similar average reward as optimal performance of model-free SAC trained for 1M timesteps (Figure 2).
>  - MPPI with true dynamics reliably solves over 90% of the problems in InhandManipulation and 100% of the problems in PegInsertion.
>  - MPQ($\lambda$) achieves similar or better performance as MPPI with true dynamics and model-free RL in the limit in all the problems.
>
> E3) The simple reward function used for CartpoleSwingup and PegInsertion were tuned by hand since we constructed these environments. The InHandManipulation environment is provided in the open-source code implementation for [1] and is available at (https://github.com/vikashplus/mj_envs) and was used without any modification. For CartpoleSwingup, states such as x-position were not normalized whereas pole-angle was kept to be between $[-\pi,\pi]$. We have added discussion on this in the Appendix in the revised draft.
>
> E4) We have updated the plots to include error bars in terms of standard error of the mean for all algorithms. We chose this metric to denote confidence over the estimated average performance of the algorithm by normalizing for finite samples. We believe this is a more representative metric than standard deviation, since it quantifies a bound on average case performance of our algorithm.

---

> > ### Author Response · Authors · 2020-11-21
> > **Response to review (2/4)**
> >
> > E5)  In Figure 3(a), (b) we study the impact of increasing levels of model degradation on the 24DOF InhandManipulation task. Here, we demonstrate that MPQ($\lambda$) can overcome a large amount of model bias and still reliably solve a highly dynamic and complex manipulation task. Further, MPQ($\lambda$) considerably outperforms base MPC that uses the biased dynamics (over 30% in terms of task success) as well as the model-free SAC baseline which lends credence to the efficacy of the approach on high dimensional tasks.
> >
> > We hope that the updated experimental evaluation sufficiently addresses the concerns of the reviewer regarding the effectiveness of MPQ($\lambda$). Next, we address the comments from the reviewer regarding the arguments in the paper.
> >
> > ### Comments
> > *************
> >
> > 1. We would like to elaborate on what we mean when we refer to MPC as a simpler, more practical alternative. First, MPC is one of the go-to approaches for highly dynamic and safety-critical tasks in robotics, including autonomous helicopter aerobatics [2], aggressive off-road driving [3, 4], legged robotics for dynamic tasks involving complex behavior generation and whole-body control [such as gymnastics on the Atlas robot by Boston Dynamics], [5, 6] and dexterous manipulation tasks involving contact [7] . In all of these tasks, **abstract models of the robot** are readily available either in the form of simplified physics or simulators. MPC can effectively leverage such models to predict the future consequences of actions before acting in the world. For example, as mentioned in the introduction, simple quasi-static models are used for tasks such as non-prehensile manipulation. Similarly, in several mobile robot navigation tasks, simple kinematic models based on Dubin’s curves are often employed. In most cases, MPC can perform these tasks in a zero-shot manner, i.e. without requiring any data. Second, while it might seem that model-free RL is a simpler approach in the sense of not requiring a dynamics model, deploying RL algorithms reliably on real-world robots in safety-critical applications is far from a solved problem. A major bottleneck is the huge amount of data required by model-free RL algorithms which can be expensive to collect on robots, and the fact that partially trained policies might perform arbitrarily poorly. Furthermore, a major advantage of MPC is to effectively account for complex non-linear constraints, something that current model-free RL approaches are not adept at. Third, it is true that MPC algorithms have certain design decisions like the horizon and optimization procedure to be used, but this issue is much more critical in model-free RL algorithms which usually have several different parameters to tune which are not as intuitive or interpretable as MPC parameters.
> >
> >     Finally, from an algorithmic and philosophical perspective, the way that the MPC paradigm attempts to solve the RL problem is more pragmatic than model-free RL. Model-free RL approaches attempt to learn a global policy that can be directly deployed at test time. This requires the class of policies considered to be expressive enough (often in the form of a deep neural network), to contain the globally optimal policy. This hurts the sample efficiency of the algorithm as searching for the policy not only requires us to solve the exploration problem to visit all relevant parts of the state space, but also requires a huge amount of data just to find the appropriate weights of the neural network using stochastic gradient descent. MPC gains in data-efficiency by focusing on simplicity in the policy class and online computation. At every timestep on the system, MPC algorithms only try to approximate the Q-function locally for a finite horizon by predicting the future consequences of actions using a simple model. By doing this repeatedly at every step, MPC has an intrinsic error correcting property that allows it to solve several tasks without relying on complex policy representations.
> >
> >   Due to the above reasons, our research focuses on using learning as a tool to improve the shortcomings of MPC to leverage our knowledge of physics, engineering, and fast optimization, rather than focusing solely on tabula-rasa learning.
> >
> >
> > 2. Thank you for recommending the relevant citations. We will include these in the final draft of the paper.

---

> > > ### Author Response · Authors · 2020-11-21
> > > **Response to review (3/4)**
> > >
> > > 3. Model bias causes systematic error in the model's prediction, and results in systematic error downstream in MPC's decisions. Model bias can thus cause the model to be inaccurate, and the amount of inaccuracy can vary across the state-action space. We have not directly measured the inaccuracy that bias produces in different areas of the environment, which could be quite complicated and difficult to report. Instead we focus on how model bias affects the task at hand: biased models result in sub-optimal decisions with different costs in different regions of the state-action space. The overall effect of model bias on decisions and the resulting task performance can be summarized by the relative difference in total cost incurred by MPC with more or less biased models. Our ablation studies on varying the amount of bias as well as comparison to MPC with access to the true model and asymptotic performance of model-free RL provide a characterization of this phenomenon. Since, unlike model-based RL, we do not use learned models but general dynamics simulators, it is not straightforward to quantify accuracy drop due to added bias globally in terms of metrics such as MSE or Negative Log Likelihood. We would like to refer the reviewer to [8] for an in-depth analysis of effects of errors in learned models on the final policy performance.
> > >
> > >
> > > 4. Our MPQ($\lambda$) framework differs from the original MPQ algorithm [9] in two major respects. First, MPQ focuses on the specific case of entropy-regularized RL, whereas our framework is more general and can be straightforwardly extended to the entropy regularized case by simply switching the Q-estimate with a soft Q-function. Second, MPQ does not take model-bias into account in a systematic manner. The learned Q-function is simply used as a terminal cost and model errors would still persist in the horizon. MPQ($\lambda$) systematically trades-off errors in both the dynamics model and learned value function by using the $\lambda$ parameter to blend multiple horizon estimates. One can view MPQ as a special case of MPQ($\lambda$) that uses soft Q-functions and $\lambda=1$. We will elaborate on this in the final draft of the paper.
> > >
> > >
> > > 5. The key takeaway of the paper is twofold. First, despite the success of Model-Predictive Control in robotics, model-bias can have catastrophic consequences in practice if not dealt with explicitly. Hand engineered models often lack the expressivity to represent reality and learned models suffer from covariate shift issues. Thus, it is indispensable for effective MPC implementations to leverage experience from the real system to mitigate the effects of such bias over time. In this paper, we provide a general framework for improving MPC with experience while systematically dealing with different sources of bias. Second, the huge sample complexity of model-free RL algorithms makes it dangerous and time consuming to deploy them on real robots in safety-critical applications. However, leveraging model predictive control can greatly improve sample complexity by incorporating prior knowledge in the form of approximate dynamics models that are readily available in robotics and engineering domains,  and exploiting online optimization with simple policy classes. We will include these arguments in the final version of the paper to make the conclusion stronger as the reviewer has suggested.
> > >
> > > 6. In this work, we have proposed a novel framework for dealing with different sources of bias in MPC while making minimal assumptions about the intricacies of the MPC algorithm. Our interpretation of MPC as cost-shaping with the value function, does not rely on an explicit terminal cost and thus, can be extended to MPC algorithms that do not have a terminal cost term in the formulation, by switching to the advantage formulation in Eq. 9.
> > >
> > > Finally, we would like to thank the reviewer for pointing out the  grammatical errors and places where language can be improved. We will fix these issues in the final draft of the paper.
> > > We hope that we have sufficiently addressed the reviewers concerns and they would consider updating their score. We would be happy to answer any remaining questions.

---

> > > > ### Author Response · Authors · 2020-11-21
> > > > **Response to review (4/4)**
> > > >
> > > >
> > > > **References**
> > > >
> > > > [1] Rajeswaran, Aravind and Kumar, Vikash and Gupta, Abhishek and Vezzani, Giulia and Schulman, John and Todorov, Emanuel and Levine, Sergey. Learning complex dexterous manipulation with deep reinforcement learning and demonstrations, 2017
> > > >
> > > > [2] Abbeel, Pieter, Adam Coates, and Andrew Y. Ng. "Autonomous helicopter aerobatics through apprenticeship learning." The International Journal of Robotics Research 29, no. 13 (2010): 1608-1639.
> > > >
> > > > [3] Williams, Grady, Nolan Wagener, Brian Goldfain, Paul Drews, James M. Rehg, Byron Boots, and Evangelos A. Theodorou. "Information theoretic MPC for model-based reinforcement learning." In 2017 IEEE International Conference on Robotics and Automation (ICRA), pp. 1714-1721. IEEE, 2017
> > > >
> > > > [4]  Williams, Grady, Paul Drews, Brian Goldfain, James M. Rehg, and Evangelos A. Theodorou. "Aggressive driving with model predictive path integral control." In 2016 IEEE International Conference on Robotics and Automation (ICRA), pp. 1433-1440. IEEE, 2016
> > > >
> > > > [5] Erez, Tom, Kendall Lowrey, Yuval Tassa, Vikash Kumar, Svetoslav Kolev, and Emanuel Todorov. "An integrated system for real-time model predictive control of humanoid robots." In 2013 13th IEEE-RAS International conference on humanoid robots (Humanoids), pp. 292-299. IEEE, 2013
> > > >
> > > > [6] Koenemann, Jonas, Andrea Del Prete, Yuval Tassa, Emanuel Todorov, Olivier Stasse, Maren Bennewitz, and Nicolas Mansard. "Whole-body model-predictive control applied to the HRP-2 humanoid." In 2015 IEEE/RSJ International Conference on Intelligent Robots and Systems (IROS), pp. 3346-3351. IEEE, 2015.
> > > >
> > > > [7] Kumar, Vikash, Emanuel Todorov, and Sergey Levine. "Optimal control with learned local models: Application to dexterous manipulation." In 2016 IEEE International Conference on Robotics and Automation (ICRA), pp. 378-383. IEEE, 2016.
> > > >
> > > >
> > > > [8] Ross, Stephane, and J. Andrew Bagnell. "Agnostic system identification for model-based reinforcement learning." arXiv preprint arXiv:1203.1007 (2012).
> > > >
> > > > [9] Bhardwaj, Mohak, Ankur Handa, Dieter Fox, and Byron Boots. "Information theoretic model predictive q-learning." In Learning for Dynamics and Control, pp. 840-850. 2020.

---

> > > ### Comment · AnonReviewer4 · 2020-11-25
> > > **reviewer response**
> > >
> > > 1. Leveraging the model could also be used for other things, such as offline RL, or designing closed form control. I agree that MPC is *generalizable*, but simpler is a tougher sell. I think there are fewer mathematical requirements, but as more people develop it, it could increase in complexity.
> > > I agree with the philosphical approach (receding horizon control is the grounding), but I think there are valid arguments for both sides.
> > >
> > > Other.
> > > You can see my other post for another reason I think the value function approach can be useful. I am curious if the authors came up with this from the point of view of a terminal cost in a traditional optimal control problem.

---

> > ### Comment · AnonReviewer4 · 2020-11-25
> > **Reviewer resposnse**
> >
> > I am still going through your responses and edits, but it takes a lot of steps to improving the paper which I appreciate.
> >
> > I had one more technical idea that may be interesting for your future work: different reward function types may benefit more from the value function embedding. Consider, for example how Cartpole has a discrete "living" reward in most settings -- MPC with a planning horizon of only a few steps may not be able to see far enough in the future to discern between actions. In this case, a value function would be extremely useful to help rank-order actions. Something like half cheetah with a more direct reward mechanism, this is not the case, though.

---

### Official Review · AnonReviewer3 · 2020-10-27
**Review for Blending MPC & Value Function Approximation for Efficient Reinforcement Learning**

**Rating:** 5
**Confidence:** 3

**Review:**


Blending MPC & Value Function Approximation for Efficient Reinforcement Learning
review:

summarization:

In this paper, the authors consider using a blending of Q value which is predicted directly from the neural network,
and a Q value which is predicted by unrolling the learnt dynamics.
The empirical results suggest improved performance,
sample efficiency and good robustness in choosing different values of blending coefficient.

Pros:
1. The idea of blending the Q values is novel.
I also think the connection to GAE is quite natural and interesting,
where both algorithms consider trade off between bias and variance (in this paper’s case, bias in learned dynamics).

2. The supporting experiments consider some of the interesting questions.
For example in section 5.1, the question of how sensitive lambda is is addressed.

3. The direction of combining value estimation in model-predictive control is interesting and under-explored.
That being said, this paper can be inspiring and helpful towards future research.

Cons:

1. The experiment section lacks comparison among state-of-the-art algorithms.
While MPPI and PPO were generally considered state-of-the-art at the time when they are published (2017),
their performance is now outperformed heavily given the fast development in the research direction.
It would be great if some of the strong baselines between 2019-2020 are included (SAC, TD3, MBPO etc.).
Also given the similarity to MCTS algorithms, it would be more convincing to include one variant of it as a baseline.

2. Experiments on more environments are also appreciated.

Questions:

I didn’t see study on the training time and testing time (how much time needed to generate one action during testing), but it seems to be referred to in the introduction?

---

> ### Author Response · Authors · 2020-11-21
> **Response to review**
>
> Thank you for investing the time to review our paper and providing valuable feedback. We address your concerns below.
>
> ### Baselines
> *************
>  - **Model Free Baselines**: We have updated the paper draft to include the Soft Actor-Critic (SAC) baseline for all environments as well as curves for average success rate for high-dimensional InHandManipulation and PegInsertion tasks. Please refer to the "Overall Response" comment for more details.
>
>
>  - **Monte-Carlo Tree Search (MCTS) Baseline**: While MCTS-based approaches have demonstrated incredible performance in discrete control problems such as game playing [1, 2], it is not straightforward to extend them to continuous control problems such as the ones studied in this paper. Although approaches such as POMCP [3] do attempt to extend MCTS to continuous state spaces, the action space is still limited to be discrete and the applicability beyond toy problems is largely unexplored. Furthermore, we would like to stress that the main aim of the work is to provide a general framework for dealing with model-bias in MPC and not to compare different MPC algorithms against each other. While it is true that a better MPC approach could potentially improve performance, an exhaustive comparison is beyond the scope of this paper. Please refer to the “Overall Response” comment for a detailed discussion.
>
>
> ### Timing Benchmark
> *********************
>
> In this work, we focused solely on the problem of mitigating the effects of different sources of bias on the performance of MPC algorithms and not on providing a highly efficient implementation for a particular MPC algorithm. Such implementations often require a significant amount of engineering effort and domain expertise. For example, in the case of sampling based controllers such as MPPI, approaches such as [4] are able to achieve real-time control by employing significant GPU acceleration along with neural network dynamics models. The choice of programming language can have a great impact on the speed of the controller as well. Our current implementation is based on Python which allows us to leverage efficient deep learning libraries such as PyTorch, but makes the controller slower compared to an implementation based on a compiled language such as C++. Since, the speed of the controller does not affect the assertions and empirical results in the current paper, we do not believe that it would be informative to include a timing analysis with the current implementation. As part of future work, we aim to deploy MPQ($\lambda$) on a real-robot platform which would warrant a more in-depth study of different engineering trade-offs involved in coming up with an efficient MPC implementation. An example of recent work that attempts to provide a nice middle-ground is [5], where the authors provide a highly efficient yet easy to use Julia framework for MPC and RL algorithms.
>
> We hope that our updated experiments and responses satisfactorily address the reviewers concerns and they would consider updating the score on that basis. We look forward to more in-depth discussions regarding the work.
>
> **References**
>
> [1] Silver, David, Aja Huang, Chris J. Maddison, Arthur Guez, Laurent Sifre, George Van Den Driessche, Julian Schrittwieser et al. "Mastering the game of Go with deep neural networks and tree search." nature 529, no. 7587 (2016): 484-489.
>
> [2] Silver, David, Thomas Hubert, Julian Schrittwieser, Ioannis Antonoglou, Matthew Lai, Arthur Guez, Marc Lanctot et al. "Mastering chess and shogi by self-play with a general reinforcement learning algorithm." arXiv preprint arXiv:1712.01815 (2017).
>
> [3] Silver, David, and Joel Veness. "Monte-Carlo planning in large POMDPs." In Advances in neural information processing systems, pp. 2164-2172. 2010.
>
> [4] Williams, Grady and Wagener, Nolan and Goldfain, Brian and Drews, Paul and Rehg, James M and Boots, Byron and Theodorou, Evangelos A. Information theoretic MPC for model-based reinforcement learning. 2017 IEEE International Conference on Robotics and Automation (ICRA)
>
> [5] Summers, Colin, Kendall Lowrey, Aravind Rajeswaran, Siddhartha Srinivasa, and Emanuel Todorov. "Lyceum: An efficient and scalable ecosystem for robot learning." arXiv preprint arXiv:2001.07343 (2020).

---

### Official Review · AnonReviewer2 · 2020-10-28
**Model based policy tuning akin to TD(\lambda) with model free base line.**

**Rating:** 7
**Confidence:** 4

**Review:**


## Summary

The paper describes an algorithm to tackle model bias in the MPC. They address the question of optimal horizon length as well the model errors observed in MPC based systems. The paper is well motivated and written in clear concise manner. Experiments with cart-pole and robot models demonstrate the practical feasibility of the proposed method.

### Strong Points

1. Good description of the sources of errors in MPC based models
2. The Theorems are useful though I was not able to check their algebraic
   accuracy, the limiting constructs are intuitively correct.
3. Choosing the horizon limit and tackling model errors are import challenges
   for MPC and the method proposed in this paper would be a good addition to the
   knowledge, hence recommendation for accept.

### To improve

1. I am not sure the MPPI is the SOTA baseline for this comparison, there are
   other MPC methods that achieve better results than MPPI.
2. Although citations from the machine learning community seem to be covered the
   standard MPC literature seem to be completely ignored. At the bare minimum,
   when speak of stability and fast horizon planning, no reference to tube based
   MPC [see Mayne 2011 ]
3. Major advantage of MPC is the ability to deal with constraints again see
   [Mayne et ref 2 below]. These are not recent developments but classic
   position papers that address lot of questions you pose and attempt to answer
   in the paper.

4. The figures need a significant improvements. In their current form the plots
   are too thin to read them correctly.



## Refs

1. Mayne, D.Q., Kerrigan, E.C., van Wyk, E.J. and Falugi, P. (2011), Tube‐based robust nonlinear model predictive control. Int. J. Robust Nonlinear Control, 21: 1341-1353. doi:10.1002/rnc.1758
2. D.Q. Mayne, J.B. Rawlings, C.V. Rao, P.O.M. Scokaert,
Constrained model predictive control: Stability and optimality,
Automatica,Volume 36, Issue 6,2000,Pages 789-814, ISSN 0005-1098,

---

> ### Author Response · Authors · 2020-11-21
> **Response to review**
>
> Thank you for reviewing our paper and providing valuable feedback. We are glad that you found the paper well motivated and clear to understand. We address your concerns below
>
> ### Citations for key MPC papers
> ***********
> We agree that the paper can be made stronger by adding in more citations to relevant papers from MPC literature such as tube MPC and other MPC approaches that focus on safety and constraints satisfaction. We will incorporate references to the papers mentioned by the reviewer as well as a more in-depth review of related MPC literature in the final version.
>
> ### Figures
> ***********
> We note the lack of clarity in the figures and have updated them in the current revision of the paper accordingly.

---

### Author Response · Authors · 2020-11-21
**Overall response to reviews (1/2)**

We would like to thank all the reviewers for investing the time to review our work and for providing helpful feedback and analysis. Here, we would like to address some common concerns raised by the reviewers, shed light on updated experimental results, and clarify some overarching themes. We hope that this will help better appreciate the work and lead to continued fruitful discussions.

### Model-free Baselines
************************

We have updated the experimental results in our paper to incorporate the Soft Actor-Critic (SAC) baseline in addition to Proximal Policy Optimization (PPO) as our model-free baseline. We use the publicly available at (https://github.com/pranz24/pytorch-soft-actor-critic), that has been used in other works in literature such as [1]. We employ automatic entropy tuning and set all other parameters set to default values from [2]. While SAC outperforms PPO on CartpoleSwingup, it does not improve over PPO in the InHandManipulation and PegInsertion environments. To the best of our knowledge, there are no previously published results of SAC successfully solving these tasks, and we found it is hard to get SAC to converge. Hence, we are currently running extensive hyperparameter searches for SAC on InHandManipulation and PegInsertion to obtain asymptotic convergence and will include these curves in the final revision. Further, PPO is able to solve both the tasks in the limit, which can be attributed to the fact that the implementation we employ (https://github.com/aravindr93/mjrl)  was  provided by the authors of [3] who also provide the open-source InHandManipulation environment (https://github.com/vikashplus/mj_envs), and has been tested extensively on such environments. Our PegInsertion environment is also based on the mjrl and mj_envs ecosystem, and hence we found it easier to get PPO to converge. These new results **further validate our claim regarding the improved sample efficiency** of MPQ($\lambda$) versus model-free RL.


### Evaluating Task Success
*****************************
To further demonstrate the ability of MPQ($\lambda$) in solving the high dimensional problems, we have included **videos of final policies** after training in the supplementary material as well as **curves for average success rate achieved by different algorithms** on the InHandManipulation and SawyerPegInsertion tasks. We provide details about the success metrics in the caption for Figure 3 and in the Appendix. In the InHandManipulation task our results clearly demonstrate that MPQ($\lambda$) with varying degrees of model degradation using bias factors of 0.6,0.8 and 1.0 (which corresponds to friction coefficients, mass and inertia being twice the true values), outperforms MPPI with a corresponding bias factor by a margin of at least 30% in terms of success rate. MPQ($\lambda$) with bias factor of 0.6 also achieves a success rate comparable to MPPI with true dynamics. Furthermore, in the PegInsertion task, even under a large standard deviation on sensor noise of 10 cm, MPQ($\lambda$) reliably solves 95-100% of the validation problems, achieving a similar level of performance to MPPI with true dynamics and model-free RL in the limit. Here, MPPI with biased dynamics is only able to solve 50% of the problems. These updated results **further solidify our claim that MPQ($\lambda$) can mitigate large amounts of model-bias in MPC in high-dimensional problems.**


### Comparison to Other MPC Optimizers
*******************************************
We would like to emphasize that the motivation of the current work is to provide a general framework for mitigating sources of bias in MPC algorithms over time by incorporating experience in the form of a learned value function. Biases introduced by incorrect models, terminal cost functions and finite horizon can be detrimental to the performance of MPC algorithms in practice and often lead to *persistent errors* in the controllers. Our proposed MPQ($\lambda$) framework tackles all sources of bias by weaving together MPC with learned value estimates. By interpreting MPC as tracing out a series of local Q-function approximations, we provide a novel method to blend these estimates with learned value estimates that is general and does not make any restrictive assumptions about the underlying MPC algorithm. Our aim here was not to compare different MPC algorithms against each other. While it is true that a better MPC approach could potentially improve performance, an exhaustive comparison is beyond the scope of this paper as it does not shed any new light on the effectiveness of MPQ($\lambda$). Given the richness of the Model-Predictive control field, we believe that such a study is a great avenue for future work. However, we kindly request that our paper be judged for the novelty and efficacy of the presented idea and not based on a problem that it doesn’t attempt to tackle.

---

> ### Author Response · Authors · 2020-11-21
> **Overall response to reviews (2/2)**
>
> ### Additional Experiments and Ablation Studies
> **************************************************
> We have also conducted additional experiments to study the robustness of MPQ($\lambda$) over the MPC horizon as well as the variance reduction properties induced by the $\lambda$ parameter when used in conjunction with sampling based MPC algorithms. The results are reported in Figures 2(f) and 2(g) in the revised draft. We make the following conclusions -  (1) decaying $\lambda$ is more robust to different decay rates and leads to better performance than tuning the discrete horizon to optimally balance model-bias (2) using $\lambda$ helps address bias-variance trade-off similar to traditional TD($\lambda$) approaches.  In MPPI it can help overcome variance issues due to limited number of particles and maintain consistent performance even over long horizons. We believe that these results further prove the efficacy of MPQ($\lambda$).
>
> We believe that our updated experiments further strengthen the claims in the paper and lend credence to the fact that MPQ($\lambda$) is a highly practical and effective framework that can have a significant impact on the fields of Model-Predictive Control and Reinforcement Learning for real-world systems. We would be happy to answer any further queries that the reviewers might have.
>
>
> **References**
>
> [1]  Okada, Masashi, and Tadahiro Taniguchi. "Variational inference mpc for bayesian model-based reinforcement learning." In Conference on Robot Learning, pp. 258-272. 2020.
>
> [2] Haarnoja, Tuomas and Zhou, Aurick and Hartikainen, Kristian and Tucker, George and Ha, Sehoon and Tan, Jie and Kumar, Vikash and Zhu, Henry and Gupta, Abhishek and Abbeel, Pieter and others. Soft actor-critic algorithms and applications, 2018
>
> [3]  Rajeswaran, Aravind and Kumar, Vikash and Gupta, Abhishek and Vezzani, Giulia and Schulman, John and Todorov, Emanuel and Levine, Sergey. Learning complex dexterous manipulation with deep reinforcement learning and demonstrations, 2017

---

### Decision · Program_Chairs · 2021-01-07
**Final Decision**

**Decision:**

Accept (Poster)

**Comment:**

The authors put a lot of effort in replying to questions and improving the paper (to a point that the reviewers felt overwhelmed).

Pros:
- An interesting way of dealing with model bias in MPC
- They successfully managed to address the most important concerns of the reviewers, with lots of additional experiments and insights
- R3's concerns have also been successfully addressed by the authors, the review & score were unfortunately not updated

Cons:
- The only remaining point is that the simulations seem to be everything but physically realistic (update at end of R1's review), which is probably a problem of the benchmarks and not the authors faults.